# LEM-3 is a midbody-tethered DNA nuclease that resolves chromatin bridges during late mitosis

Ye Hong[1,2], Remi Sonneville[2], Bin Wang[1], Viktor Scheidt[1], Bettina Meier[1], Alexander Woglar[3,4], Sarah Demetriou[1], Karim Labib[2], Verena Jantsch[3] & Anton Gartner[1]

Faithful chromosome segregation and genome maintenance requires the removal of all DNA bridges that physically link chromosomes before cells divide. Using *C. elegans* embryos we show that the LEM-3/Ankle1 nuclease defines a previously undescribed genome integrity mechanism by processing DNA bridges right before cells divide. LEM-3 acts at the midbody, the structure where abscission occurs at the end of cytokinesis. LEM-3 localization depends on factors needed for midbody assembly, and LEM-3 accumulation is increased and prolonged when chromatin bridges are trapped at the cleavage plane. LEM-3 locally processes chromatin bridges that arise from incomplete DNA replication, unresolved recombination intermediates, or the perturbance of chromosome structure. Proper LEM-3 midbody localization and function is regulated by AIR-2/Aurora B kinase. Strikingly, LEM-3 acts cooperatively with the BRC-1/BRCA1 homologous recombination factor to promote genome integrity. These findings provide a molecular basis for the suspected role of the LEM-3 orthologue Ankle1 in human breast cancer.

[1] Centre for Gene Regulation and Expression, School of Life Sciences, University of Dundee, Dundee, DD1 5EH, UK. [2] MRC Protein Phosphorylation and Ubiquitylation Unit, School of Life Sciences, University of Dundee, Dundee, DD1 5EH, UK. [3] Department of Chromosome Biology, Max F. Perutz Laboratories, Vienna Biocenter, University of Vienna, Vienna, A-1030, Austria. [4] Present address: Departments of Developmental Biology and Genetics, Stanford University School of Medicine, Stanford, CA 94305-5329, USA. Correspondence and requests for materials should be addressed to A.G. (email: a.gartner@dundee.ac.uk)

Faithful chromosome segregation requires all connections that physically link chromosomes to be removed before cells divide; failure to do so may lead to the severing of chromosomes, prevent cell division or cause aneuploidy and polyploidization[1–4]. It is widely known that proteinaceous connections between chromosomes mediated by cohesins must be removed prior to segregation, but so must DNA structures that link chromatids. These DNA connections include DNA repair intermediates, points at which chromatids have become intertwined, and loci that have not fully replicated. Cytologically, connections between separating chromatids can take the form of chromatin bridges or ultrafine DNA bridges[5,6]. The majority of DNA linkages, such as branched recombination intermediates, are "dissolved" by the combined action of the BLM helicase and Topoisomerase III concomitant with DNA replication, or by the MUS81-EME1 nuclease in G2/M and the GEN1 Holliday junction resolvase during anaphase (for review ref. [7]). However, remaining chromatin bridges can persist into telophase. During cytokinesis the Aurora B kinase dependent NoCut checkpoint detects chromatin bridges and delays abscission of daughter cells in order to allow for chromatin bridge resolution[4,8]. This delay is achieved through continuous Aurora B activation, whereas cytokinesis is completed upon kinase inactivation. However, little is known about how chromatin bridges are resolved when the Aurora B kinase delays cytokinesis.

The LEM-3 nuclease was discovered in a genetic screen for DNA repair genes in *C. elegans*[9]. *lem-3* mutants are hypersensitive to ionizing radiation (IR), UV light and DNA cross-linking agents[9]. However, previous studies have not shown how LEM-3 mediates genome stability. The mammalian orthologue of LEM-3 is known as Ankle1 and is predominantly expressed in hematopoietic tissues. Ankle1-deficient mice are viable and show no detectable defects[10–12] Interestingly, non-coding polymorphisms in either the ANKLE1 locus or the neighboring ABHD8 locus have been associated with increased risk of breast and ovarian cancer in the general population and in carriers of BRCA1 mutations, suggesting that altered expression of ANKLE1 or ABHD8 might contribute to tumourigenesis[13–15].

Here we identify a novel mechanism conferred by LEM-3 nuclease to preserve genome integrity during the final stages of cell division in *C. elegans*. Our data indicates that LEM-3 can be regulated by the Aurora B kinase AIR-2 and provides a final safeguard to resolve the remaining chromatin bridges during late mitosis, by virtue of the recruitment of LEM-3 to the midbody.

## Results

**LEM-3 acts outside of known DNA repair pathways**. *C. elegans lem-3* null mutants were reported to proliferate normally but are hypersensitive to a range of DNA damaging agents including ionizing irradiation (IR), which predominantly acts by inducing DNA double-strand breaks[9]. The mammalian orthologue of LEM-3, known as Ankle1, is poorly characterized[10–12]. We wished to determine whether LEM-3 acts in previously characterized DNA damage repair pathways, or else defines a new mechanism. We found that after IR treatment, mutation of *lem-3* showed more severe phenotypes when combined with null alleles defective in the three major DNA double-strand break repair modalities: BRC-1-dependent homologous recombination (HR), polymerase Theta (POLQ)-mediated end-joining[16], and LIG-4-dependent non-homologous end-joining[17] (Supplementary Figures 1a, c). Taken together, our genetic data indicate that LEM-3 might act in a previously unknown response mechanism.

**LEM-3 accumulates at the midpoint of chromatin bridges**. To better understand how LEM-3 helps to maintain genome stability,

we assessed LEM-3 localization using an existing transgene carrying a *lem-3::YFP* translational fusion, as well as a GFP tagged *lem-3* derivate, generated by CRISPR-Cas9-mediated genome editing. Both constructs complement the DNA damage hypersensitivity phenotypes conferred by *lem-3* null alleles (Supplementary Figure 6a)[9]. Consistent with previous findings[9], LEM-3 appeared excluded from the nucleus, and formed prominent foci of uncertain nature outside the nuclei in the absence of DNA damage (Fig. 1a). To explore potential links between these LEM-3 foci and chromosome segregation, we exposed *C. elegans* to a range of treatments that induce chromatin bridges during cell division. As shown in Fig. 1b, LEM-3 localized to the center of chromatin bridges that formed between dividing nuclei in response to DNA breaks (IR), DNA replication defects (hydroxyurea treatment or RNAi, targeting the MCM-7 helicase subunit), condensation defects (RNAi, to deplete the HCP-6 condensin II subunit), or decatenation defects (RNAi depletion of the type II topoisomerase TOP-2[18,19]). Careful examination of LEM-3 localization by OMX super-resolution microscopy of DAPI stained embryos treated with *mcm-7* RNAi revealed that LEM-3 initially formed a ring structure encircling chromatin bridges between two separated daughter cells (Fig. 1c; Supplementary Figure 7d). In telophase nuclei, this structure shrinks accompanied by the gradual elongation of chromatin bridges. Finally, LEM-3 concentrates in the center of chromatin bridges between late telophase nuclei (Fig. 1c; Supplementary Figure 7d).

**LEM-3 is a midbody-tethered protein**. Given the LEM-3 staining pattern, we speculated that LEM-3 might localize at the midbody, which defines the site of cell abscission, the complete separation of two daughter cells at the end of cytokinesis. Using a strain carrying YFP-LEM-3 and mCherry-histone H2B fusions we found that YFP-LEM-3 localizes to the midbody region starting from telophase (Fig. 1d, white arrows; Supplementary Movie 1). To establish whether LEM-3 indeed co-localizes with the midbody, we compared the relative localization of LEM-3 and ZEN-4. ZEN-4 is a component of the centralspindlin complex, comprises the ZEN-4/MKLP1 kinesin -6 motor protein and the CYK-4 Rho family GAP[20], both essential for midbody formation and cytokinesis. We observed that LEM-3 co-localizes with ZEN-4 at the midbody (Fig. 2a; Supplementary Movie 2). Furthermore, midbody localization of LEM-3 could not be observed upon RNAi depletion of CYK-4 or SPD-1, proteins essential for the integrity of the midbody, even in the presence of chromatin bridges induced by *mcm-7* RNAi (Fig. 2b, c). We also investigated if LEM-3 localization required cleavage furrow ingression. As expected, depletion of a contractile ring component NMY-2 inhibited furrow formation and ingression but did not affect the central spindle formation (Supplementary Figure 1d). We found that LEM-3 could still be detected at the midzone upon *nmy-2* RNAi (Supplementary Figure 1d). In summary, the LEM-3 nuclease associates with the midbody, and LEM-3 midbody location depends on the formation of the central spindle and the midbody.

**LEM-3 is required for the resolution of chromatin bridges**. We next sought to determine whether LEM-3 is required to resolve chromatin bridges that result from incomplete DNA replication, defective chromosome condensation, or unresolved recombination intermediates. We found that LEM-3 started to accumulate earlier at the midbody, and the intensity of LEM-3 foci was stronger in the presence of chromatin bridges generated by *mcm-7* RNAi or the condensin I subunit *capg-1* RNAi (Fig. 1d–h, white arrows for LEM-3, white arrowheads for chromatin bridges; Supplementary Movie 3).

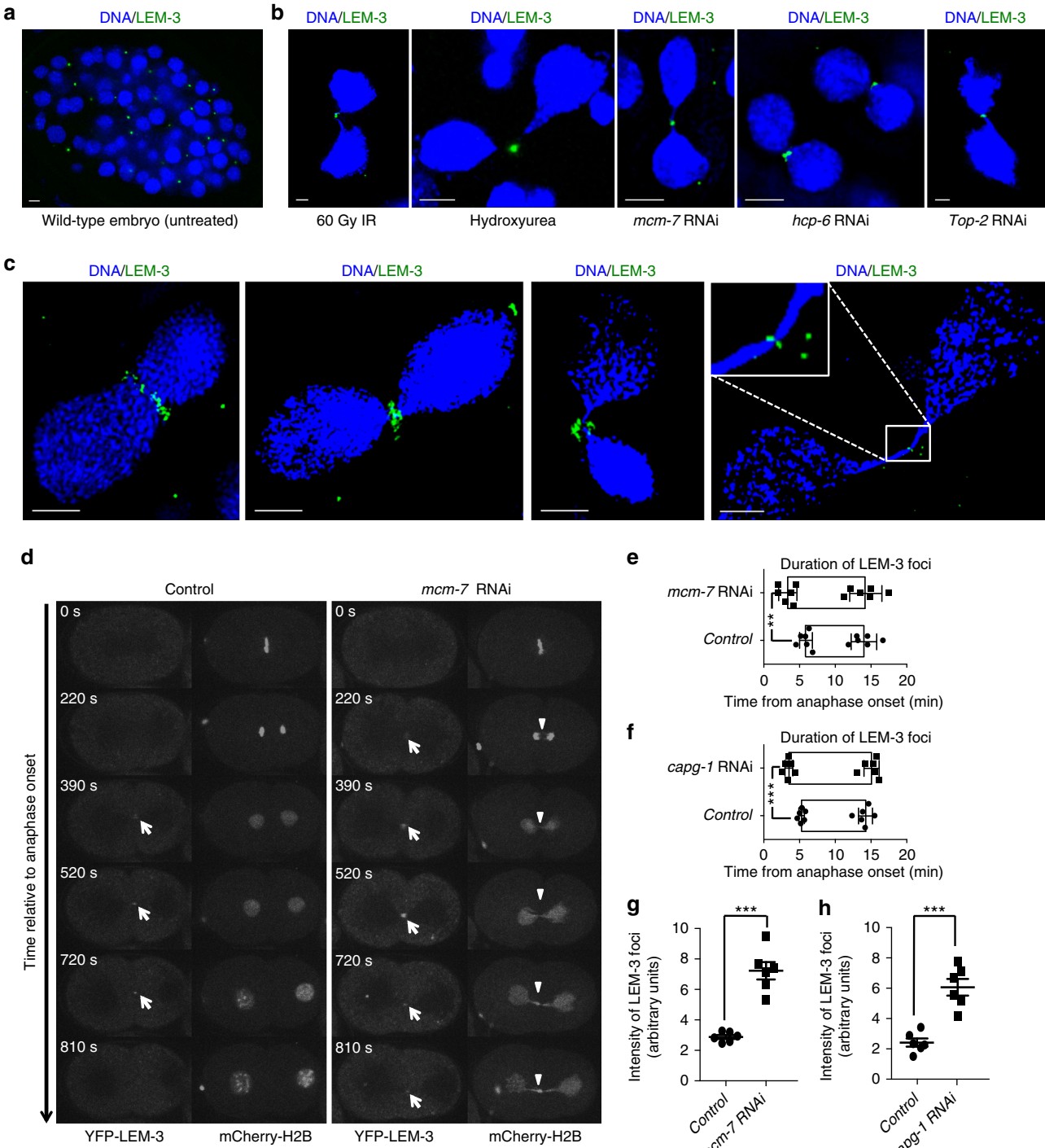

**Fig. 1** Localization of LEM-3 on chromatin bridges and at the midbody. **a** Localization of YFP-LEM-3 in wild-type embryo. **b** Localization of YFP-LEM-3 on chromatin bridges generated by IR, hydroxyurea, and RNAi depletion of replicative helicase subunit MCM-7, condensin II component HCP-6 and topoisomerase TOP-2. **c** Representative OMX images of YFP-LEM-3 localization at different stage of chromosome segregation in the presence of chromatin bridges induced by *mcm-7* RNAi. Scale bars: 2 μm. **d** Localization of LEM-3 on chromatin bridges induced by *mcm-7* RNAi. Arrows indicate YFP-LEM-3. Arrowheads indicate the chromatin bridges. Times are relative to anaphase onset of the first division. **e**, **f** Quantification of the average time duration of GFP-LEM-3 localization at the midzone/midbody during the first mitotic cell division in wild type, *mcm-7* RNAi (**e**) and *capg-1* RNAi (**f**) embryos. Error bars represent standard deviation of the mean. **g**, **h** Quantification of the intensity of GFP-LEM-3 foci during the first mitotic division in wild type, *mcm-7* RNAi (**g**) (measured at 2.5 min after first appearance of LEM-3 foci) and *capg-1* RNAi (**h**) embryos. Asterisks indicate statistical significance as determined by two-tailed Student *t*-test. *p* values below 0.05 were consider significant, where *p* < 0.05 was indicated with *, *p* < 0.01 with **, and *p* < 0.001 with ***

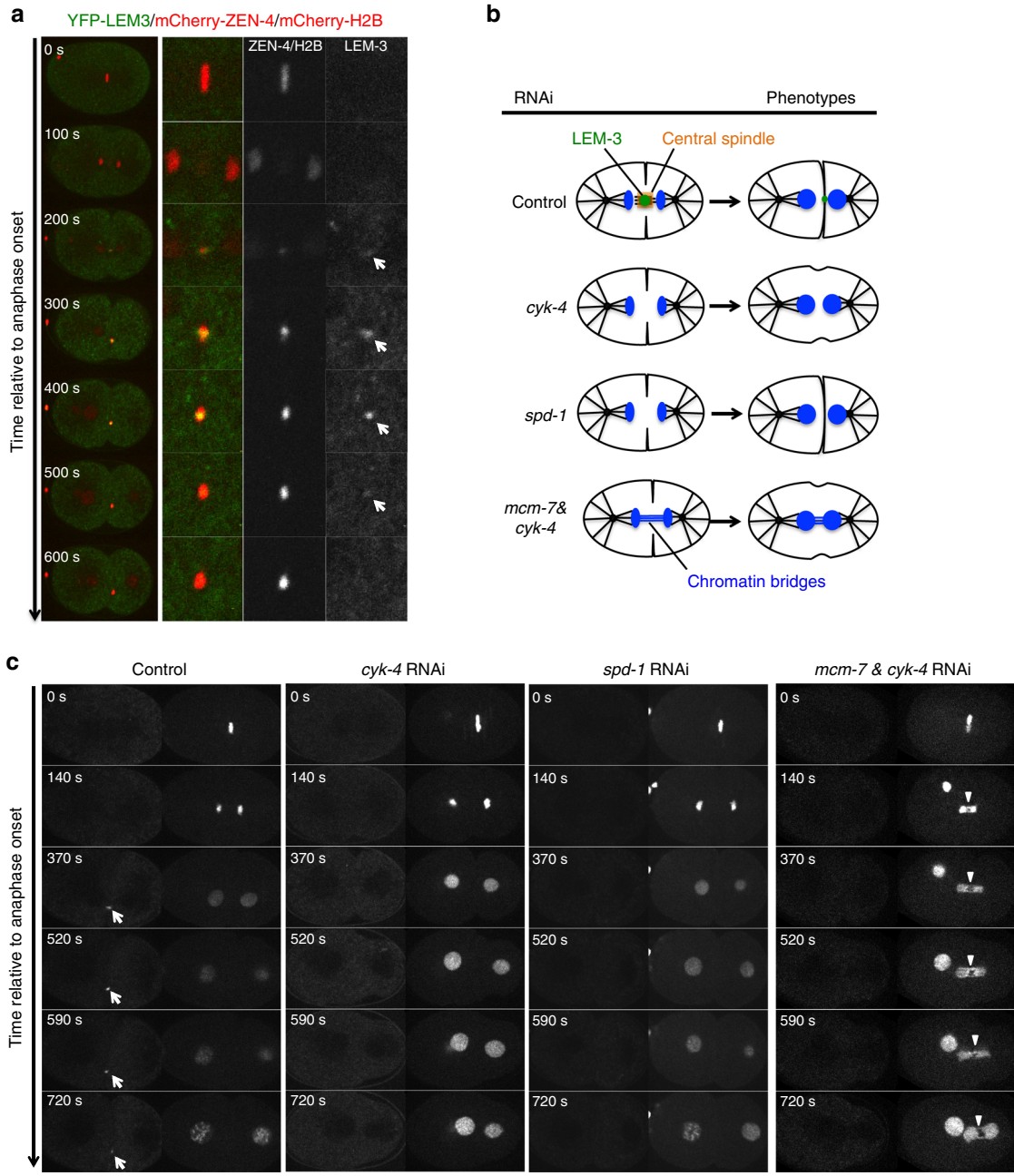

**Fig. 2** Central spindle formation is required for midbody localization of LEM-3. **a** LEM-3 co-localized with the central spindle component ZEN-4. Arrows indicate YFP-LEM-3. **b** Schematic of LEM-3 localization in control embryos (normal central spindle formation) and *cyk-4*, *spd-1* RNAi embryos (defective central spindle formation). **c** Midbody localization of LEM-3 depends on the central spindle component CYK-4 and the microtubule-bundling protein SPD-1. Depletion of the central spindle by *cyk-4* RNAi, no LEM-3 foci are detectable even in the presence of chromatin bridges generated by *mcm-7* RNAi. Images were taken from a time-lapse recording of the *cyk-4* RNAi, *spd-1* RNAi, and *cyk-4* combined with *mcm-7* RNAi-treated embryos expressing YFP-LEM-3 and mCherry-H2B from the anaphase onset of the first mitotic division. Arrowheads indicate chromatin bridges. Times are relative to anaphase onset

When DNA replication was mildly impaired by weak depletion of the MCM-7 helicase subunit via low dose RNAi (10% bacteria expressing *mcm-7* RNAi), cell cycle progression occurred normally, and no chromatin bridges were observed (Fig. 3a; 10/10 embryos), consistent with our previous work[18]. In contrast, chromatin bridges formed and persisted into the next cell division in all cases (12/12 embryos; Fig. 3a) in *lem-3* mutants. In the absence of replication stress inflicted by *mcm-7* depletion no chromatin bridges formed in *lem-3* mutants (Supplementary Figures 2b and 5c; 10/10 embryos). It thus appears that DNA bridges arising from incomplete DNA replication are efficiently processed in wild type, but remain unresolved in *lem-3* mutants. Consistent with this view, we found that chromatin bridges were eventually resolved at the end of the first cell division in 33% of wild-type embryos exposed to the 100% *mcm-7* RNAi ($n = 9$), whereas bridges persisted into the next division in all *lem-3* mutant embryos treated with 100% *mcm-7* RNAi ($n = 14$) (Fig. 3b). Similarly, while the chromatin bridges induced by depletion of the Condensin I subunit CAPG-1 were eventually resolved in wild type ($n = 5$), all *lem-3* embryos tested showed persistent unresolved bridges ($n = 5$) (Fig. 3c, white arrowheads).

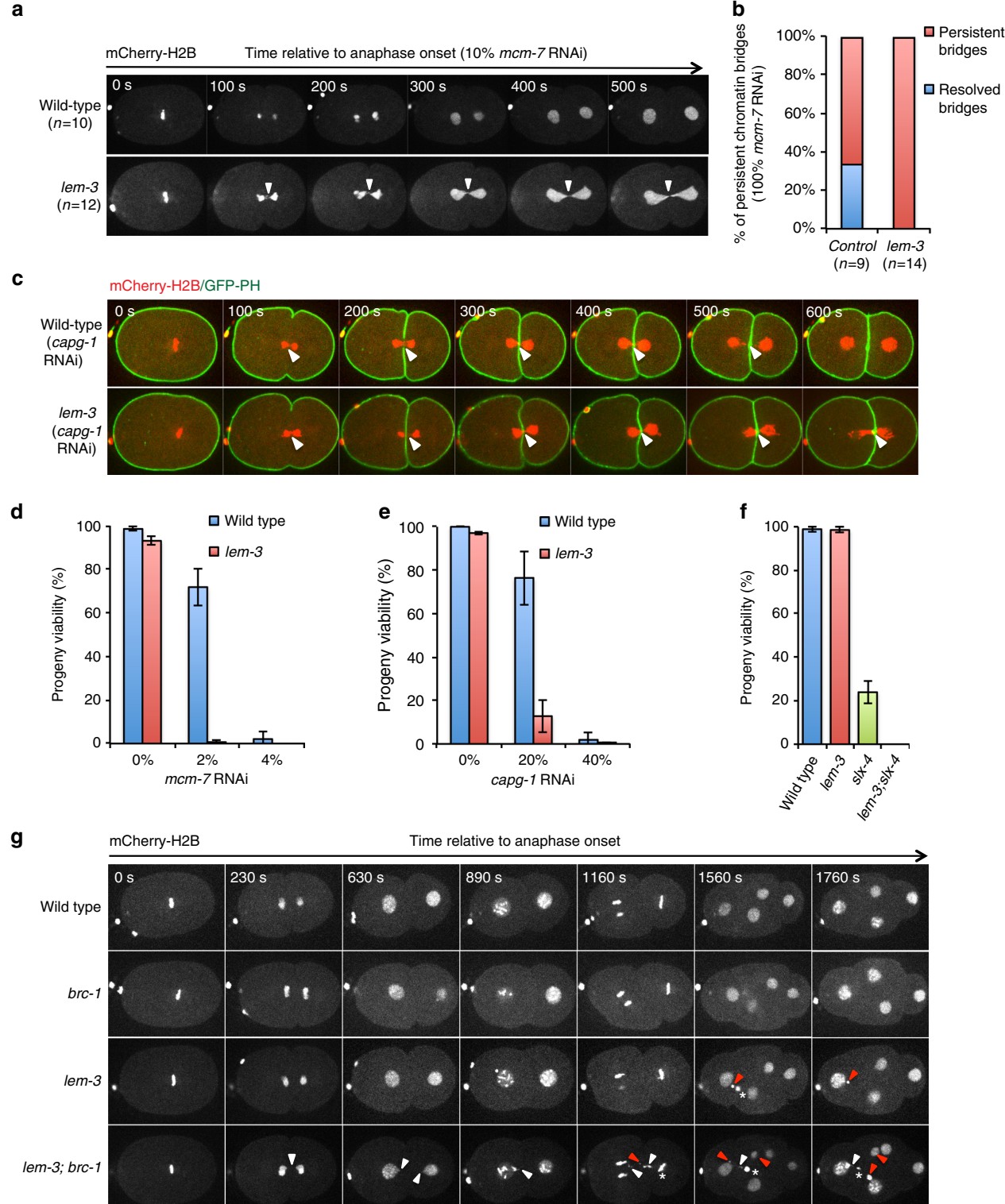

**Fig. 3** LEM-3 contributes to chromatin-bridge resolution. **a** Chromosome segregation in wild type and *lem-3* mutant embryos upon partial depletion of DNA replication helicase subunit MCM-7. Images were taken from embryos expressing mCherry-H2B fed on 10% *mcm-7* RNAi producing bacteria. Arrowheads indicate chromatin bridges. Times are relative to anaphase onset of the first division. **b** Quantification of chromatin bridges that persist into the next cell division in wild type and *lem-3* mutant embryos upon full depletion of MCM-7 (100% *mcm-7* RNAi). *n* sample size. **c** Persistent *capg-1* RNAi induced chromatin bridges in *lem-3* mutants. Chromosome segregation in wild type and *lem-3* mutant embryos upon depletion of condensin I subunit CAPG-1 (100% *capg-1* RNAi). Images were taken from control and *capg-1* RNAi embryos expressing mCherry-H2B and GFP-PH. **d**, **e** Progeny viability of *lem-3* mutants upon partial depletion of MCM-7 (**d**) and CAPG-1 (**e**). **f** Genetic interaction between LEM-3 and SLX-4. Error bars represent standard deviation of the mean. **g** LEM-3 and BRC-1 are required for faithful chromosome segregation after IR. White arrowheads indicate chromosome bridges, red arrowheads indicate micronuclei. Stars indicate polar bodies

To further explore the functional significance of chromatin-bridge resolution by LEM-3, we examined embryonic viability in either *lem-3* mutants alone or in combination with non-lethal low-doses RNAi against *mcm-7* or *capg-1*. Whereas 72 and 76% of wild type *C. elegans* survived partial RNAi depletion with *mcm-7* RNAi (2% bacteria expressing *mcm-7* RNAi) or *capg-1* RNAi (20% bacteria expressing *capg-1* RNAi) respectively, the same RNAi treatment led to only 0.6% and 12.8% viable embryos in *lem-3* mutants (Fig. 3d, e). The hypersensitivity of *lem-3* mutants to the partial inhibition of MCM-7 and CAPG-1 indicates that LEM-3 is important to process DNA intermediates that result from incomplete DNA replication or partial chromosome decondensation to ensure viability. We then investigated whether the same was true for DNA bridges that result from incomplete processing of DNA damage. As described above, *lem-3; brc-1* double mutants showed increased sensitivity to IR compared to the single mutants (Supplementary Figure 1a). This loss of viability in the double mutant upon IR treatment is associated with extensive chromatin-bridge formation, and lagging chromosomes and micronuclei became apparent, especially upon real-time imaging (Fig. 3g, Supplementary Figure 2b for control, Supplementary Movie 4). It thus appears that LEM-3 is important to process DNA bridges that result from the incomplete processing of DNA breaks when BRC-1 function is compromised.

Other endonucleases such as MUS-81 and SLX-1-SLX-4 have been shown to resolve recombination intermediates during metaphase (for review ref. [7]). To investigate the contribution of LEM-3 to recombination intermediates processing in relation to these other endonucleases, we assessed the progeny viability of *lem-3* and *slx-4* single and double mutants. SLX-4 scaffold protein interacts with MUS-81 and SLX-1 and provides a platform for recombination intermediate resolution. In *slx-4* mutants unresolved meiotic recombination intermediates are carried into the first zygotic division[21,22]. Although *slx-4* single mutants still produce 20% viable embryos, *lem-3; slx-4* double mutants only produce inviable progeny (Fig. 3f). Imaging of the first zygotic cell division showed that chromatin bridges did occur in *slx-4* embryos, but were barely visible. They had no apparent effect on chromosome segregation based on distance measurements between separating nuclei, indicating that the chromatin bridges can be efficiently resolved (Supplementary Figures 5a and c). In contrast, chromosome segregation was severely compromised in *lem-3; slx-4* embryos, as indicated by the defective separation of nuclei and the formation of chromatin bridges and micronuclei (Supplementary Figures 5a, c, and d). Interestingly, the separation of nuclei occurred faster in *lem-3* mutants with average nuclear separation rate of 5.31 μm/100 s (standard deviation SD: ±0.13; $n = 5$), compared to 4.02 μm/100 s in wild type (SD: ±0.32; $n = 6$, $p < 0.0001$), 3.77 μm/100 s in *slx-4* mutants (SD: ±0.23; $n = 6$, $p < 0.00001$), and 2.4 μm/100 s in *lem-3; slx-4* double mutants (SD: ±0.37; $n = 5$, $p < 0.00001$), indicating a potential role for LEM-3 in delaying the separation of nuclei (Supplementary Figure 5a). Careful analysis revealed that in 32% of *lem-3; slx-4* embryos ($n = 19$) cytokinesis failed, leading to binucleated cells and highly abnormal mitotic cell divisions (Supplementary Figures 5b and d). In summary, our data suggest that LEM-3 processes recombination intermediates, as well as DNA structures resulting from replication stress or partial chromosome decondensation, and that such processing helps to maintain genome integrity.

**Bridge resolution requires LEM-3 midzone/midbody localization**. To better correlate the spatiotemporal behavior of chromatin-bridge resolution and the LEM-3 midbody localization, we simultaneously visualized the localization of YFP-LEM-3,

the plasma membrane (mCherry-PH) and chromatin (mCherry-H2B) when chromatin bridges are induced by *capg-1* RNAi. We found that LEM-3 was first detectable at the midzone at the time of cleavage furrow completion, then increased in abundance and together with the chromatin bridges congressed into the midbody (Supplementary Figure 2c and d, white arrow; Supplementary Movie 5). At that stage the bridges started to resolve and to retract, commencing from the midbody (Supplementary Figure 2c white arrowheads, 810–830 s), consistent with the function of midbody-tethered LEM-3.

To further test whether the ability of LEM-3 to resolve chromosome bridges requires an intact spindle midzone and the recruitment of LEM-3 to that site, we combined the partial *mcm-7* RNAi with the depletion of ZEN-4 and CYK-4 central spindle components. As expected we found that compromising central spindle formation abolished midzone localization of LEM-3 (Fig. 4; Supplementary Figure 3). Embryos solely treated with low dose of *mcm-7* RNAi never showed formation of chromatin bridges, as was the case for *zen-4* or *cyk-4* RNAi-treated embryos (Fig. 4; Supplementary Figure 3). In contrast, when embryos treated with combination of partial *mcm-7* RNAi and *zen-4* or *cyk-4* RNAi, chromatin bridges formed with high penetrance in (4/7) and (4/9) cases, respectively (Fig. 4; Supplementary Figure 3; Supplementary Movies 6 and 7). This finding indicates that the spindle midzone is important for the resolution of chromatin bridges, and by inference suggests that LEM-3 midbody localization might have a role in this.

**Midbody localization of LEM-3 requires the AIR-2/Aurora B kinase**. The Aurora B kinase dependent NoCut checkpoint is able to detect chromatin bridges and delay abscission, allowing time for chromatin-bridge resolution[4,8,19]. We found that the *C. elegans* Aurora B homolog AIR-2 and LEM-3 partially co-localize at the midzone/midbody. Using Manders' overlay coefficient 77.4% ± 9.3% of LEM-3 protein co-localized with AIR-2 at the midzone/midbody (Fig. 5a). In addition, LEM-3 localization depends on the AIR-2/Aurora B kinase (Supplementary Figure 4). This could be because AIR-2 is required for midbody formation[23], which in turn is required for LEM-3 localization (Fig. 2c). Alternatively, we consider that AIR-2 might directly regulate the midbody association and/or activity of LEM-3. Searching for AIR-2/Aurora B consensus phosphorylation sites (K/R; K/R; X0-2; S/T) in the LEM-3 protein sequence we identified two serines, Ser192 and Ser194, embedded in such a consensus sequence (Fig. 5b). The corresponding sites are phosphorylated by AIR-2/Aurora B in ZEN-4/MKLP4, and the first serine also occurs in human LEM-3 orthologue Ankle1[24]. We thus mutated serines 192 and 194 to alanine by genome engineering (Fig. 5b). We observed comparable expression levels of GFP-LEM-3 S192A S194A and the wild-type control (GFP-LEM-3, Fig. 5c). However, the *lem-3 S192A S194A* mutant showed an increased sensitivity to IR upon treatment of L4 stage larvae and late stage embryos (Fig. 5d; Supplementary Figure 6a). In addition, combining the *lem-3 S192A S194A* mutations with *slx-4* resulted in reduced viability compared to *slx-4* mutant, consistent with increased levels of chromatin bridges observed in *lem-3 S192A S194A; slx-4* double mutants (Supplementary Figures 6b and c). Careful examination of the localization of LEM-3 revealed no detectable LEM-3 foci at the midzone in the *GFP-LEM-3 S192A S194A* mutant during the first cell division (Fig. 5e; Supplementary Movie 8). Upon condensin inactivation by *capg-1* RNAi, GFP-LEM-3 S192A S194A midbody location appeared later and reduced as compared to wild type (Fig. 5e, white arrows; Supplementary Movie 9). Consistent with LEM-3 midbody location contributing to chromatin resolution, chromatin bridges

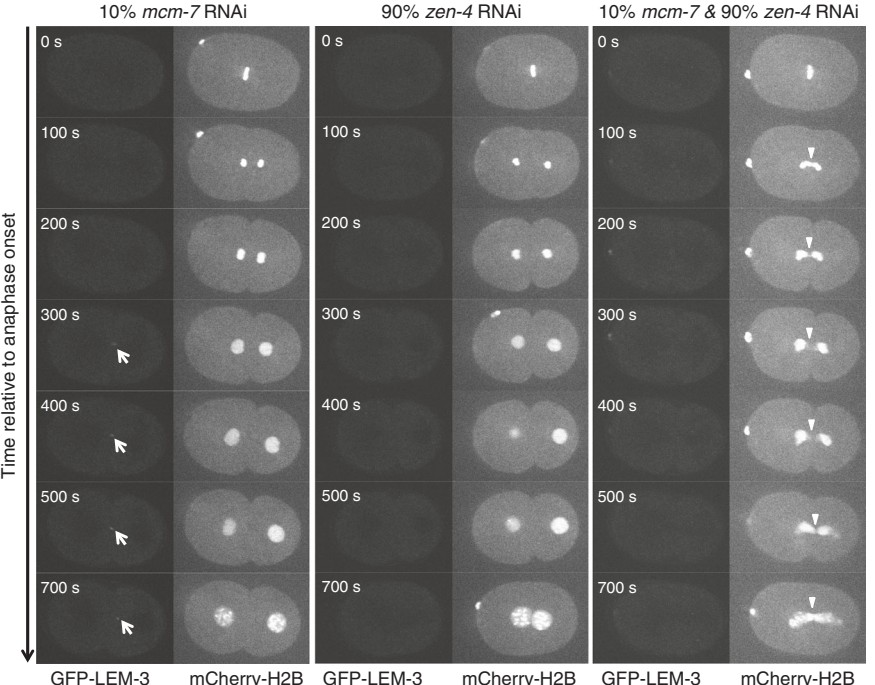

**Fig. 4** Formation of chromatin bridges upon partial depletion of MCM-7 and central spindle component ZEN-4. Arrowheads indicate chromatin bridges

persisted longer when the LEM-3 phosphorylation mutant embryos were treated with *capg-1* RNAi (Fig. 5e, white arrowheads).

To provide direct evidence that LEM-3 S192 and S194 are phosphorylated in vivo, we generated phospho-specific antibodies corresponding to a peptide containing both phosphorylated S192 and S194 (Supplementary Figure 7a). Using these LEM-3 phospho-specific antibodies we detected LEM-3 midbody-staining that overlapped with the localization of GFP-LEM-3 (Fig. 6a). The same co-localization occurred on chromatin bridges induced by *capg-1* and *mcm-7* RNAi (Fig. 6b; Supplementary Figure 7d). Importantly, no LEM-3 phosphorylation was detected on the midbody, when *lem-3 S192A S194A* and *lem-3 (tm3468)* mutants (the latter containing an in-frame deletion that abrogates phosphorylation sites) were stained with the phospho-specific antibody (Supplementary Figures 7b and c). Whereas GFP-LEM-3 is detected between nuclei from anaphase onwards in the presence of chromatin bridges induced by *capg-1* RNAi, phosphorylated LEM-3 is only detected at the midbody during telophase (Fig. 6b). Therefore phosphorylation of LEM-3 might not be required for the initial recruitment but for congression of LEM-3 into the midbody. In summary, our data indicate that AIR-2 is required for robust localization of LEM-3, and LEM-3 phosphorylation may be important for stable association of LEM-3 with the midbody.

**The GIY-YIG motif is essential for in vivo function of LEM-3.** To determine whether the conserved GIY-YIG nuclease motif is required for LEM-3 function, we mutated the catalytic Y556 and G558 residues, which are predicted to be essential for nuclease activity[25,26]. Interestingly, LEM-3 localization at the midzone/midbody was not detectable in the *Y556A G558A* double mutant even after induction of chromatin bridges by *capg-1* RNAi (Fig. 5e; Supplementary Movie 10) suggesting that the mutations we introduced into the GIY-YIG motif might also compromise DNA binding and or midbody localization. *lem-3 Y556A G558A* has the same level of protein expression as wild type but is hypersensitive to IR (Fig. 5c, d; Supplementary Figure 6a).

Moreover, combining the *lem-3 Y556A G558A* mutation with *slx-4* led to 100% embryonic lethality and extensive chromatin-bridge formation (Supplementary Figures 6b and c). In addition, excessive and persistent chromatin-bridge formation occurred upon *capg-1* RNAi in *lem-3 Y556A G558A* embryos (Fig. 5e; Supplementary Movie 10). These results suggest that the GIY-YIG motif is essential for LEM-3 function in vivo.

## Discussion

It has been known for a decade that the Aurora B-mediated NoCut checkpoint has a key role in delaying abscission, when chromatin is trapped at the midbody during cytokinesis[4,8]. However, it remained enigmatic how chromatin-bridge resolution occurs and how this process is connected with the abscission machinery and the NoCut checkpoint. In this study, we report that the conserved endonuclease LEM-3 accumulates at the midbody and provide evidence that it resolves chromatin bridges resulting from multiple perturbations of DNA metabolism at the final stages of cell division during anaphase and cytokinesis (Fig. 6c). LEM-3 midbody localization requires central spindle formation and is likely to be regulated by the AIR-2/Aurora B kinase (Fig. 6c). Together with our evidence that LEM-3 does not act within the known major DNA repair pathways, this suggests that LEM-3 defines a new mechanism for maintaining genome integrity that acts at the late stage of mitosis to resolve chromatin bridges.

Our data indicate that Aurora B promotes bridge resolution via recruitment of LEM-3 to the midbody. We found that the *C. elegans* Aurora B kinase AIR-2 co-localizes with LEM-3 and that the depletion of AIR-2 abolished the association of LEM-3 with the midbody (Fig. 5a; Supplementary Figure 4b). This could be an indirect effect, as AIR-2 is required for midbody formation[23]. Although we have no evidence for the direct interaction between LEM-3 and AIR-2, we favor the hypothesis that AIR-2 directly regulates LEM-3. LEM-3 has conserved AIR-2/Aurora B phosphorylation sites and these can be phosphorylated in vivo. Thus, LEM-3 may be a substrate of AIR-2/Aurora B kinase, in line with the co-localization of both proteins. The occurrence of phosphorylated LEM-3 during cytokinesis is consistent with the

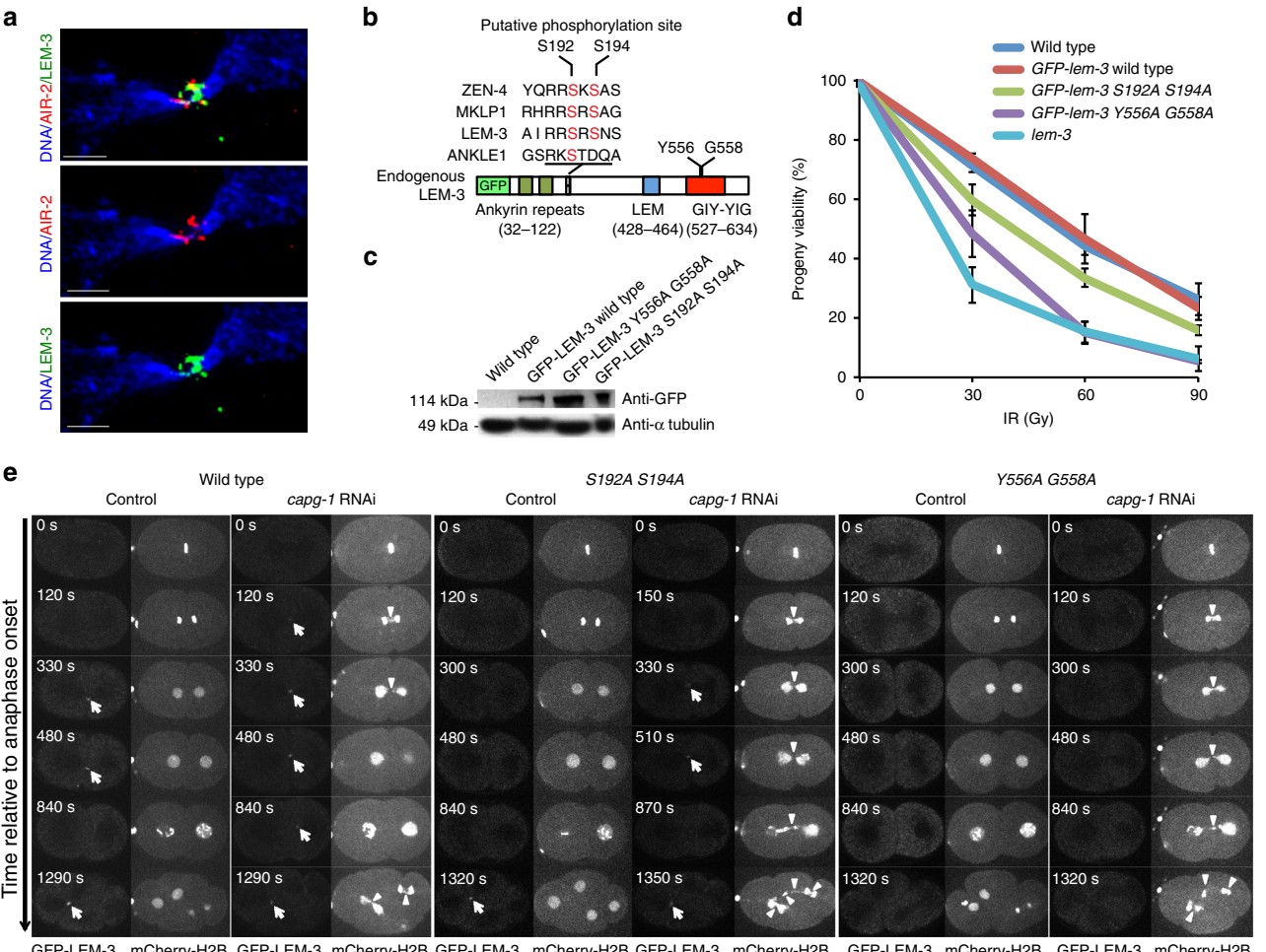

**Fig. 5** Functional dissection of LEM-3 phosphorylation sites and GIY-YIG motif. **a** Partial co-localization of LEM-3 with AIR-2/Aurora B kinase. Co-localization was quantified by using ImageJ to calculate Manders' overlap coefficient (MOC); 77.4% (standard deviation, ±9.3%; $n = 12$) of the LEM-3 protein co-localized with AIR-2. AIR-2 immunostaining of representative embryos expressing GFP-LEM-3. Scale bars: 1 μm. **b** Schematic presentation of endogenous GFP-LEM-3 generated by CRISPR/Cas9-mediated genome editing. The conserved Serine 192 and 194 AIR-2 putative phosphorylation sites and Tyrosine 556 and Glycine 558 in the GIY-YIG motif are indicated. **c** Western blot analysis of GFP-LEM-3 protein levels in GFP-LEM-3 wild type, *GFP-lem-3 S192A S194A* and *GFP-lem-3 Y556A G558A* mutants. **d** Sensitivity of *GFP-lem-3 S192A S194A* and *GFP-lem-3 Y556A G558A* mutants to ionizing radiation (IR). Progeny viability of *lem-3* mutants after young adult nematodes were treated with IR as described in Supplementary Figure 1a. Values are the average of three independent experiments. Error bars represent standard deviation of the mean. **e** Analysis of chromosome segregation in *GFP-lem-3 S192A S194A* and *GFP-lem-3 Y556A G558A* mutants. Arrows indicate GFP-LEM-3. Arrowheads indicate the chromatin bridges

reported high level of AIR-2 kinase activity at this stage[19]. In agreement with the hypothesis of midbody-tethered LEM-3 processing DNA intermediates during cytokinesis bridges started to resolve and to retract, commencing from the midbody upon depletion of the condensin I component CAPG-1 (Supplementary Figure 2c). Our finding that midzone/midbody formation has a key role in chromatin-bridge resolution further supports the functional importance of LEM-3 midbody localization (Fig. 4; Supplementary Figure 3). It is noteworthy that the *lem-3 S192A S194A* phosphosite mutant is not as sensitive to IR as the *lem-3* null mutant. Thus, mutation of these two serines impairs but does not completely abolish LEM-3 function (Fig. 5d, e; Supplementary Figure 6a). This may be due to the residual midbody location of LEM-3 in this mutant (Fig. 5e). LEM-3 localizes to the midzone prior to being phosphorylated at the S192A S194A sites (Fig. 6b). Thus LEM-3 might not only act at the midbody during cytokinesis but also at the midzone during anaphase. A role in anaphase could explain that we did not find anaphase brides in late anaphase upon partial *mcm-7* RNAi while bridges were observed when the same *mcm-7* depletion was conducted in

ZEN-4 and CYK-4 depleted embryos (Fig. 4; Supplementary Figure 3), suggesting that LEM-3 already starts to act during anaphase. The *C. elegans* checkpoint preferentially works after the second embryonic cell cycle[19], rendering chromatin-bridge processing by LEM-3 more important in the first cell cycle. Indeed, the faster separation of nuclei we observed in *lem-3* mutants may suggest that LEM-3 also has a direct role in the NoCut checkpoint. Irrespectively, while we provide strong evidence that LEM-3 acts on the midzone/midbody during late mitosis, from anaphase to cytokinesis, we cannot rule out that LEM-3 might also have a role in earlier stages of the cell cycle.

We found that LEM-3 is able to bind to and resolve chromatin bridges caused by various perturbations, including incomplete DNA replication, unresolved recombination intermediates, and compromised chromosome condensation (Figs. 1b and 3a, c; Supplementary Figure 5), indicating that LEM-3 can process a wide range of DNA substrates with distinct structures. In organisms with large genome, unreplicated DNA is frequently present in normal cells at the end of G2 phase[27]. The post-replicative resolution of unreplicated DNA depends on creation

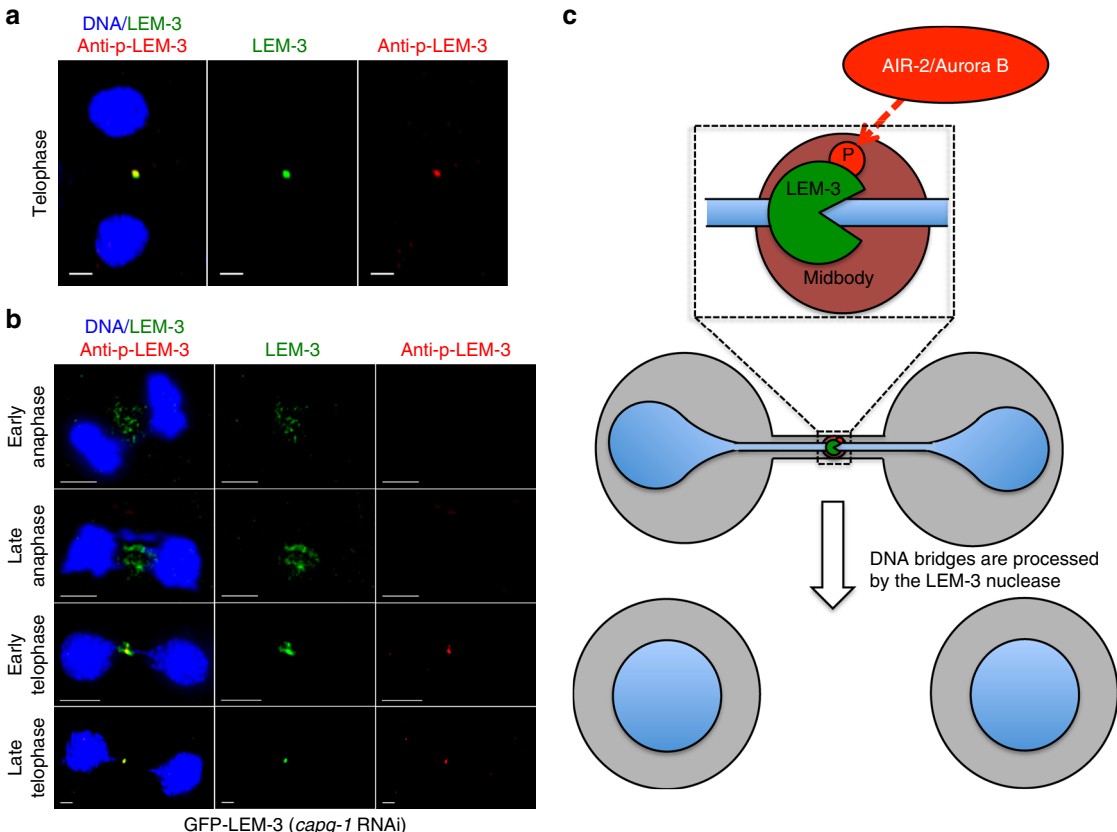

**Fig. 6** LEM-3 is phosphorylated at the midbody. **a** Co-localization of GFP-LEM-3 and phospho-LEM-3 at the midbody. Scale bars: 5 μm. **b** Localization of LEM-3 and phosphorylated LEM-3 in the presence of chromatin bridges from anaphase to telophase. **c** Proposed model for chromatin-bridge processing by the midbody associated LEM-3 nuclease

of ultrafine anaphase bridges during mitosis[27]. Ultrafine DNA bridges, which cannot be stained by conventional DNA dyes are different from chromatin bridges[28]. It is possible that the ultrafine DNA bridges exist in early *C. elegans* embryos where rapid cell divisions occur. Ultra fine bridges in mammalian cells are detected by being coated with the BLM or the PICH helicase[29–31]. We did not observe bridges coated with the *C. elegans* BLM ortholog HIM-6, and PICH has not been found encoded in the nematode genome (our unpublished data). Irrespective, in order to segregate unreplicated segments of DNA, unreplicated double stranded DNA has to be disentangled to form single stranded DNA, which can be repaired in the next cell cycle[27]. Therefore, the accumulation of LEM-3 at the midbody during a late stage of cell division may also contribute to the resolution of the ultrafine DNA bridges caused by unreplicated DNA regions in unperturbed cells.

In human cells, the cytoplasmic nuclease TREX1 was recently found to contribute to the cutting of chromatin bridges that originate from telomere fusion[32]. TREX1 binds to the chromatin bridges after anaphase, and its nuclease activity is important for their timely cleavage. Unlike LEM-3, whose localization becomes more restricted at the midbody as cytokinesis proceeds, TREX1 is present throughout chromatin bridges and generate extensive ssDNA. Moreover, TREX1 is an exonuclease that requires nicked DNA substrates, indicating that TREX1 is not solely responsible for chromatin-bridge resolution, or else can only act on nicked DNA substrates. Furthermore, in contrast to LEM-3, which processes chromatin bridges at the end of cell division to maintain genome stability and allow for increased survival, this is not clear for TREX1, which might only get access to bridges when the nuclear envelope is perturbed. In addition, the cutting of

telomere-telomere bridges by TREX1 leads to pathological "McClintock-like" breakage-fusion-bridge cycles often linked with a large number of randomly assembled chromosome fragments, a phenomenon termed Chromothripsis[32–34]. In contrast, our data suggest that LEM-3 is able to process a variety of DNA bridges to promote viability (Fig. 3d, e). Thus LEM-3 is part of a "last chance saloon" mechanism to maintain genome integrity.

It will be very interesting in future studies to explore further the role of the Ankle1 nuclease[10], which is the ortholog of *C. elegans* LEM-3 in human and mouse. The localization of endogenous Ankle1 has neither been reported, nor its potential association with chromatin bridges explored, though over-expressed Ankle1 protein was found to be cytoplasmic[10]. Mice lacking Ankle1 do not have an overt phenotype and extracted cells are not hypersensitive to DNA damaging agents[11], but this is also true of mice lacking the Gen1 or Mus81 nucleases, indicating a high level of redundancy between DNA nucleases that preserve genome integrity in mammalian systems[35]. Interestingly, increased susceptibility to breast and ovarian cancer in the general population and in carriers of BRCA1 and BRCA2 mutations was mapped down to 13 polymorphisms at 19p13.1 mapping to non-coding regions in the human LEM-3 homolog Ankle1 and a second gene ABHD8, encoding for a alpha beta hydrolase domain[13–15]. The effect of these polymorphisms on Ankle1 function remains to be explored, but our finding that *lem-3; brc-1* double mutants show elevated sensitivity to DNA damage (Fig. 3g; Supplementary Figure 1a) is consistent with Ankle1 being the most likely gene in 19p13.1 to affect breast cancer disposition. Indeed, another study found that coding mutations in Ankle1 were associated with increased risk of breast cancer[36]. It will be important in the future to explore whether Ankle1 and

BRCA1 cooperatively promote genome stability in human cells, as predicted by the behavior of their *C. elegans* equivalents.

## Methods

**Experimental model and subject details**. *C. elegans* strains were maintained at 20 °C on nematode growth medium (NGM) plates seeded with OP50 bacteria. Translational fusion lines were generated by injecting plasmid DNA/ssDNA directly into the hermaphrodite gonad. Integrated GFP-*lem-3* strains were outcrossed at least six times before use. The list of strain is provided in the Supplementary Table 1.

**Embryonic viability assay**. L4 or young adult nematodes were treated with various DNA damaging agents as previously described[37]. After 24 h of recovery, nematodes were transferred onto new plates and allowed to lay eggs for 6–8 h. Eggs were quantified. Unhatched eggs were counted 24 h later and the percentage of progeny viability was calculated.

**Embryo irradiation assay**. The embryo irradiation assays[17,37]. Experiments were done in triplicate. 10 young adult nematodes of each strain were transferred onto seeded plates. Nematodes were allowed to lay eggs for 2 h and were removed from the plate afterwards. The embryos were incubated at 20 °C for 3 h before treatment with different doses of IR (0 Gy, 30 Gy, 60 Gy, 90 Gy). The eggs were counted, and the plates were incubated at 20 °C for 24 h. Unhatched eggs were counted the next day and plates were incubated for another 24 h. The delayed development phenotype was scored 48 h after IR by calculating the percentage of nematodes that reached the L4 stage.

**Image acquisition**. For immunostaining, embryos were fixed in methanol at −20 °C for 30 min. After three washes with PBST (0.1% Tween 20), samples were blocked with 0.1% BSA, incubated with primary antibodies overnight, and washed three times. Following a 2-h incubation with secondary antibodies and/or DAPI, cover glasses were rinsed five times before image analysis. Primary and secondary antibodies were used at the indicated dilutions: rabbit anti-AIR-2 (1:200)[38], sheep anti-phospho-LEM-3 (1:100), Alexa 568-labeled donkey anti-rabbit (1:750), and Alexa 594-labeled donkey anti-sheep (1:1000) (Invitrogen). For super-resolution OMX microscopy, established protocols were followed[39]. Embryos were dissected in M9 (3 g/l KH$_2$PO$_4$, 6 g/l Na$_2$HPO$_4$, 5 g/l NaCl, 1 mM MgSO$_4$) and mounted on 2% agarose pads. Images were captured every 10 s at 23–24 °C using a spinning-disk confocal microscope (IX81; Olympus) with spinning-disk head (CSU-X1; Yokogawa Electric Corporation) and MetaMorph software (Molecular Devices)[18]. Image analysis and video processing were performed using ImageJ software (National Institutes of Health).

**RNAi**. Clones for *mcm-7*, *capg-1*, *top-2*, *nmy-2*, *spd-1*, *zen-4*, and *cyk-4* feeding strains were obtained from Source BioScience. To make RNAi plates, bacteria were grown to OD600 0.6–0.8. After induction with 1 mM IPTG for 4 h at 37 °C, bacteria were spread on NGM plates. After 16 h L4 nematodes were transferred to RNAi plates and fed for at least 24 h at 25 °C before analysis. For double RNAi, equal amounts of bacterial cultures were mixed prior to seeding the RNAi plates. Partial *mcm-7* and *capg-1* RNAi were achieved by diluting *mcm-7* or *capg-1* RNAi expressing bacteria with control bacteria[18].

**Generation of point mutations by CRISPR/Cas9 genome editing**. The *cop859* [P*lem-3*::eGFP::STag:: *lem-3*::3′UTR*lem-3*] eGFP insertion was generated by Knudra (http://www.knudra.com/) following the procedures described by Dickinson et al.[40] Exact details are available upon request. For generated point mutations in the endogenous *lem-3* locus, we performed CRISPR-Cas9 following procedures described by the Seydoux lab[41] using modified protocol provided by Simone Köhler and Abby Dernburg. The injection mixture was prepared by pre-assembling gRNA-Cas9 ribonucleoprotein (RNP) complexes in vitro. 1:1 mixture of tracrRNA and crRNA (100 μM each) were heated at 95 °C for 5 min, followed by annealing at room temperature for 5 min. Cas9-RNP complex was then formed by adding purified Cas9 protein (IDT) into a injection mixture (10 μl) with a final concentration of 30 μM gRNA and 28 μM Cas9 protein and incubated at room temperature for 20 min. Repair templates were provided as 100 ng/μl ssDNA. 2.5 ng/μl of pCFJ90 and 5 ng/μl of pCFJ104 plasmids were used as red fluorescent co-injection makers. 10–40 young adult hermaphrodites were injected for each mutant and recovered onto individual plate. Fluorescent F1 nematodes were picked after 2–3 days post-injection and screened by PCR or DNA sequencing. crRNA sequence for GFP-*lem-3S192A S194A* was 5′-AAGAAGTAGATCCAACAGCA-3′, and the repair template sequence (ssDNA, Ultramer oligo, IDT) was as follows: cggttcaatcgaggaaatgttcttactaattcatatagatgtgcaaagaagaaaatcagagccacatttcatgctattcgaag aGCtagaGcTaacagcacCgcaacacttcaagacgttgtttttaacatctgaaggaattcgaacagtgaccacacct agtaggagagcacctaaggcaaccgtttatgc.

crRNA sequence for GFP-*lem-3 Y556A Y558A* was 5′-GTACGATCAATCTT TTACGT-3′, and the repair template sequence (ssDNA, Ultramer oligo, IDT) was ttcggatataatgcgttttgctatctcattatggatcctcgaattttgggaagcaatgtggagaaccttacacttgaaac

ctttgtacgatcGatctttGCcgtagCTaaaggttcaaaaaatcgtccattagctcatttcattgatgctcgcaatgaacg gagagacaaattggataaactaaaaacttgcgaaaag.

**Western blotting**. Wild type, GFP-*lem-3*, GFP-*lem-3 S192A S194A*, and GFP-*lem-3 Y556A Y558A* adult nematodes were collected and bleached until they were completely lysed. After centrifugation for 1 min at 2200 rpm the pellet was resuspended in 200 μl "M9 worm buffer". The suspension was transferred into safety capped 1.5 ml reaction tubes. Overall, 200 μl of 40% TCA and 300 μl of glass beads were added and the nematode eggs were disrupted by bead beating for 2 min. The suspension was then transferred into a 1.5 ml reaction tube. The beads were washed with 5% TCA, which was added to the suspension afterwards. Samples were spun down for 10 min at 3000 rpm, the supernatant was removed and the pellet was resuspended in 1 ml ice-cold acetone. After another centrifugation for 2 min at 11,000 rpm the acetone was discarded and the pellet was left to dry at room temperature. Finally the pellet was resuspended in 100 μl Laemmli buffer (3×). Overall, 10 μl of each sample was loaded onto a 4–12% NuPage precast gel. For Western blotting primary and secondary antibodies were used at the indicated dilutions: mouse anti-GFP (1:1000), mouse anti-α-tubulin (1:5000), horse anti-mouse IgG-HRP (1:3000) (NEB).

**Data availability**. All relevant data are available from the authors.

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

## Acknowledgements

This work was funded by Wellcome Trust Programme (AG 0909444/Z/09/Z), Investigator (KL 102943/Z/13/Z) and Strategic awards (097045/B/11/Z), a MRC core grant KL (MC_UU_12016/13), and the FWF (VJ SFB-F34). V.S. and Y.H. were supported by a Wellcome PhD fellowship and ISSF funds. The Dundee Imaging Facility was supported by the Wellcome Trust (097945/B/11/Z) and the MRC (MR/K015869/1). We thank the Caenorhabditis Genetics Center (funded by NIH Office of Research Infrastructure Programs P40 OD010440) for *C. elegans* strains. We thank Federico Pelisch for AIR-2 antibodies, James Hastie and Fiona Brown for phospho-specific LEM-3 antibodies (MRC PPU reagents, https://mrcppureagents.dundee.ac.uk). We are grateful to Graeme Ball, Callum Tromans-Coia and Markus Posch for technical assistance. We thank Simone Köhler and Abby Dernburg for sharing genome-engineering protocols. We thank Ulrike Gartner for proofreading.

## Author contributions

Experiments were mostly done by Y.H., with contributions from R.S., B.W., V.S., B.M., and A.W. Y.H., A.G., V.J., and K.L. wrote the paper.

## Additional information

**Competing interests:** The authors declare no competing financial interests.

