## [Peer Review File · Nature Communications]

Reviewers' comments:

Reviewer #1 (Remarks to the Author):

"LEM-3 is a midbody-tethered DNA nuclease that resolves chromatin bridges during cytokinesis"
by Hong et al

This study examines the role of the DNA nuclease LEM-3 in chromosome segregation and cytokinesis during the first division in the *C. elegans* embryo. LEM-3 was originally identified in a screen for radiation-sensitive mutants but its function remained unclear. Hong and colleagues find that LEM-3 localizes to the midbody region, and that this localization is more pronounced in cells with chromatin bridges caused by irradiation and by defects in DNA replication, condensation or decatenation. The midbody localization of LEM-3 requires the central spindle and Aurora-B kinase function. LEM-3 is phosphorylated in Aurora-B consensus sites, and mutation of these sites leads to radiation sensitivity and LEM-3 localization defects, suggesting that Aurora-B phosphorylation promotes LEM-3 function and its recruitment to the midbody. Finally, mutation of LEM-3 inhibits the resolution of chromatin bridges which normally occurs during cytokinesis. The authors conclude that LEM-3 acts at the midbody to promote the resolution of replication or recombination intermediates and that this promotes genome stability.

The observations reported in this study are of high interest, and provide valuable insight into the function of LEM-3, its role in promoting chromosome segregation, and more generally into how DNA bridges are processed during cytokinesis. However, I don't agree that the main conclusion of the paper, that LEM-3 resolves bridges during cytokinesis, is well supported by the data. I suggest that the authors revise their conclusions or provide direct evidence for their claims. There are also a number of relatively minor issues that it would be good to address before publication, as outlined below.

1. The evidence that LEM-3 resolves chromatin bridges during cytokinesis is correlative: LEM-3 is shown at the midbody at the time of chromatin bridge resolution (Fig. 3F, which is very nice) and *lem3* mutants show longer lived bridges (Fig. 3B and S2, also technically beautiful). It is therefore tempting to speculate that LEM-3 cleaves bridges during cytokinesis. However, stable bridges in *lem-3* mutants may have alternative causes. Lack of LEM-3 in mitosis or in the previous S-phase could cause changes in the quantity or quality of bridge DNA, making these bridges more stable and/or causing abscission to be inhibited. Thus, midbody recruitment of LEM-3 and bridge resolution may be, in principle, functionally unrelated events. Unless direct evidence is provided (such as inactivation of LEM-3 specifically during cytokinesis), the authors should make clear in the title, abstract and main text that the evidence supporting LEM-3 dependent cleavage of bridges at the midbody is indirect and that alternative interpretations cannot be excluded.

2. The authors claim that LEM-3 function at the midbody is important to preserve genome integrity (Discussion, 1st paragraph and also later). Again, no direct evidence supports the claim that LEM-3 functions during cytokinesis. Even if LEM-3 preserves genome integrity (which is an inference), it could do so through a function in previous cell cycle stages, not necessarily depending on its midbody localization during cytokinesis.

3. As noted in the Discussion, nuclease recruitment to chromatin bridges might be deleterious to genome stability, by causing chromosome fragmentation. The authors should comment on how they think LEM-3 processing of chromatin bridges could differ from other nucleases such as TREX1 to preserve genome integrity. In the specific case of unreplicated bridges, the authors may want to discuss their data in the context of a DNA resolution model proposed by Julian Blow and collaborators (Moreno et al., PNAS 2016, doi: 10.1073/pnas.1603252113).

4. More thorough quantification of the observed phenotypes would strengthen the observed correlations between bridge resolution and LEM-3 function. For instance: In Fig 3B, S2, 5D, how

much longer do LEM-3 deficient bridges persist, compared to controls? In Fig S4B, by how much is nuclear separation advanced in *lem-3* mutants? In Fig. 1E-F, is advanced recruitment of LEM-3 to the midbody statistically significant?

5. The midbody localization of LEM-3 requires the central spindle and Aurora-B kinase function (Figure 4). Is a midbody formed in the absence of AIR-2 function? If not, the role of AIR-2 in LEM-3 localization in these experiments could be indirect. This should be noted. To directly test the role of Aurora-B in LEM-3 midbody recruitment, the authors could inactivate AIR-2 after anaphase onset, using an *air-2* ts mutant such as *air-2(or207)*.

6. "The Aurora B kinase dependent NoCut checkpoint is essential to detect chromatin bridges and prevent abscission until chromatin bridges have been properly resolved" (p. 6, bottom). At least in budding yeast, NoCut delays abscission but does not prevent resolution of decondensed and catenated bridges (Amaral et al., Nat Cell Biol 2016, doi: 10.1038/ncb3343).

7. The localizations of LEM-3 and AIR-2 at the midbody seem mutually exclusive in some images (Fig 4A). This should be quantified.

8. LEM-3 localizes as a broad band bisecting the segregating chromosomes in Figure 1C. Could this be the contractile ring, or the spindle midzone microtubules? This localization is not seen in other figures, where LEM-3 only is visible at the midbody. Is this due to differences in imaging techniques?

9. "Unlike LEM-3, TREX1 associates throughout chromatin bridges and not just with the region at the midbody" (p. 9). However, LEM-3 also seems to associate with chromatin bridges during anaphase (see Fig. 6B).

Reviewer #2 (Remarks to the Author):

This manuscript makes a convincing case that LEM-3 acts to promote accurate chromosome segregation in the early *C. elegans* embryo. The authors present evidence for enhancement of a number of apparent null alleles in DNA repair pathways, consistent with the idea that it acts in a novel repair pathway. The authors also convincingly demonstrate that LEM-3 accumulates at the midbody and they suggest that it functions at the location. However, the evidence for this latter conclusion is not convincing and further analysis along these lines is warranted prior to publication. This also renders the title problematic.

The central conclusion of the manuscript that LEM-3 acts at the midbody is based on LEM-3 protein localization and the dependence of this localization on central spindle and Aurora B. This conclusion has several problems. First, no evidence is shown that the ability of LEM-3 to resolve chromosome bridges requires an intact spindle midzone and LEM-3 recruitment to that site. This could be easily tested by combining some of the weak DNA damage perturbations with ZEN-4 depletion and scoring for an enhancement of the frequency of chromosome bridges. Second, on page 6, the authors state " In addition, LEM-3 localization depends on the AIR-2/Aurora B Kinase (Fig. 4B), indicating that AIR-2 regulates the midbody association and/or activity of LEM-3"

Given that AIR-2 is required for spindle midzone assembly and the previously demonstrated dependence of LEM-3 localization on ZEN-4 (though not necessarily function - see above), this is a trivial result.

- In figure 1A, untreated embryos accumulate LEM-3 at midbodies independent of DNA damage. This should be explicitly stated. Interestingly, this accumulation appears to depend on S192S194 phosphorylation and catalytic activity (Figure 5). While this suggests it is not strictly non specific, given that it is damage independent, it is not clear why the mutations in the catalytic site would impair this damage independent localization. How do the authors explain this result? Does the p-LEM-3 antibody detect the epitope in control embryos that do not express GFP-LEM-3?
- Page 4: LEM-3 "dynamically" localizes to the midzone. The data does not show dynamic localization. This would require FRAP or related technique to show protein dynamics. Regulated accumulation is more accurate.
- The evidence that Aurora B phosphorylation promotes LEM-3 function is not highly convincing. The difference in radiation resistance is not particularly impaired relative to the wild-type, it only mildly enhances the inviability in combination with slx-4 mutation. The part of the manuscript might be better de-emphasized.
- The authors use operational terms for the extent of RNAi mediated depletion, corresponding to dilution of the bacteria used for RNAi. Given that there is no quantification of extent of depletion, these numbers are meaningless to the reader. They should use qualitative terms like partial, or weak.
- In some of the experiments (see page 4), the number of the analyzed embryos is on the low side. 10 embryos is more frequently used in the field.
- Figure preparation: Where appropriate, all datapoints should be shown in bar graphs, e.g. Fig E-H. It is also convention that embryos are oriented with the anterior to the left, not vertically.
- The authors should point out in the main text that the him-18, hrc-1, brc-1, and lem-3 alleles are nulls. This is critical for the conclusion that LEM-3 acts in a distinct pathway.

Minor points mostly involving unartful phrasing:

- Page 2, "failure to do so can lead to the severing of chromosomes, prevent cell division or cause aneuploidy and polyploidization 1-4, molecular pathologies that all result in disease." This is stated too strongly and is therefore incorrect. Many chromosome segregation defects result in death of the affected cell and no subsequent disease. Certainly there are other cases that are disease-associated. In addition, the processes that affect repair of some chromosome segregation defects can improve situation but they do not necessarily ensure euploidy. For example, post anaphase, perhaps the repair events prevent chromosome breakage, but the daughter cell genomes are not necessarily intact.
- Page 2, "next to nothing", perhaps little is better.
- Page 4, "If CYK-4 or SPD-1, a protein". Incorrect tense.
- Page 5, "Whereas 80% of wild type worms survived treatment with 2% of mcm-7 RNAi or 20% of cpag-1 RNAi respectively, either treatment led to a strong synthetic lethal effect in lem-3 mutants (Fig. 3C and 3D)." State the result quantitatively, with more specificity than strong.
- Page 6, "found that LEM-3 localization commenced". LEM-3 was first detectable at the midzone at the time of cleavage furrow completion.

- Page 6, "The corresponding sites are known to be". Awkward phrasing.

- Page 10, last chance 'saloon'. Inappropriate placement of quotes. Should be "last chance saloon".

Reviewer #3 (Remarks to the Author):

The authors make a strong case that LEM-3 acts at the midbody of dividing cells to resolve DNA bridges. First, the combination of mutations in *lem-3* with mutations in genes involved in any of the three major DSB repair pathways leads to elevated IR hypersensitivity. Second, LEM-3 localizes to the midbody of dividing cells. Third, in *lem-3* mutants, bridges resulting from DNA damage or replication stress are apparently not resolved. The authors also show convincing evidence that LEM-3 has two phosphorylation sites that may facilitate its regulation by AIR-2 kinase, the *C. elegans* Aurora B homolog. These results will be interesting to the DNA repair field as they suggest a new mechanism for the resolution of chromatin bridges.

Major concerns

1. Introduction: It would be helpful to describe what is known about LEM-3 and Ankle1 rather than summarizing three prior publications with "Ankle1 is poorly characterized."

2. The first two sections of results: "LEM-3 accumulates at the midpoint of chromatin bridges during cell division" and "LEM-3 associates with the midbody during cytokinesis" are similar titles, but the former has a more broad characterization of LEM-3 and *lem-3* mutants. These sections need to be organized more clearly.

3. The authors state "*lem-3* mutants synergize with mutants defective in the three major DNA double-strand break repair modalities," but it is unclear that this figure shows synergy in the mutants. Synergism in genetic terms means that the effect seen in the double mutant is greater than the combination of the two single mutants (that is, more than just an additive effect). Fig. S1A doesn't appear to show synergy between *brc-1* and *lem-3*. Although one would expect a synergistic increase when mutations in two different pathways are combined, if the effect is really additive it needn't argue against their hypothesis. Finally, a "mutant" is an individual, whereas a "mutation" is a change in the genetic material. It is mutations that are showing additive effects (in double-mutant animals).

4. Page 4 – "We observed LEM-3 localization at the centre of the midbody slightly after ZEN-4 midbody location became apparent (Fig. 1I; Movie S3)." I do not see clear evidence of LEM-3 appearance after ZEN-4, it appears it is at the same time according to Fig. 1I and Movie S3.

5. Figure 3A. An important control to show here is *lem-3* mutants without the 10% *mcm-7* RNAi to show that the chromatin bridges are a result of having replication stress, and not just the *lem-3* mutation.

6. Page 6- "separation of nuclei occurs faster in *lem-3* mutants, indicating a potential role for LEM-3 in delaying the separation of nuclei (Fig. S4B)." This figure also shows that the *lem-3*; *slx-4* double mutants have a slower separation of nuclei compared to the control, which is the opposite phenotype of the *lem-3* single mutants. The authors should comment this observation?

7. Page 6- Figure 3F. The authors say there is an increase of abundance of LEM-3 localization, but this is unclear from the figure and is not quantified. It is also unclear from the text that this is in a

capg-1 RNAi background.

Minor concerns

1. Figure 1C- include a label showing this is mcm-7 RNAi. Are these pictures from a time course? If so, label it as such.
2. Figure 2A. In the mcm-7 & cyk-4 mutant, are the lines between the nuclei supposed to represent chromatin bridges? Because they look very similar to the central spindle in the control. Perhaps color the DNA, so it is clear that those lines are DNA and not the central spindle.
3. Figure 1E-H are never referenced to in the text. This could belong with the first paragraph of the section, "LEM-3 is required for the resolution of chromatin bridges" pages 4/5.
4. Page 9- second paragraph of Discussion. Could also note that the phosphosite mutant is not as affected by IR as a lem-3 null, suggesting that the residual midbody localization of the phosphosite mutant could be partially rescuing this phenotype.

Reviewer #1 (Remarks to the Author):

We were glad to see that this reviewer judged that **“the observations reported in this study are of high interest, and provide valuable insight into the function of LEM-3, its role in promoting chromosome segregation, and more generally into how DNA bridges are processed during cytokinesis.”**

The reviewer then raised one major and several minor points, as follows:

Major point:

“I don't agree that the main conclusion of the paper, that LEM-3 resolves bridges during cytokinesis, is well supported by the data. I suggest that the authors revise their conclusions or provide direct evidence for their claims.”

Our response: We thank the reviewer for some very helpful comments and have made major changes to our manuscript as detailed below, both in the form of new data and by making textual changes. Most importantly, we now provide further evidence that LEM-3 acts at the midbody to resolve chromatin bridges, by including an experiment that was suggested by reviewer 2. We show that depletion of the central spindle components ZEN-4 or CYK-4, which are required for localization of LEM-3 during cytokinesis, leads to increased frequency of chromosome bridges. As shown in Figures 3F and S3B, embryos solely treated with low dose of *mcm-7* RNAi, or with RNAi against *zen-4* or *cyk-4*, never showed chromatin bridges. In contrast, when embryos were treated with a combination of partial *mcm-7* RNAi and *zen-4* or *cyk-4* RNAi, chromatin bridges formed with a high penetrance in (4/7) and (4/9) cases, respectively (Fig. 3F and S3B, Movies S6 and S7). These results demonstrate that midzone assembly is required for chromatin bridge resolution, as well as for LEM-3 recruitment and cytokinesis completion.

Minor points:

1. **“The evidence that LEM-3 resolves chromatin bridges during cytokinesis is correlative: LEM-3 is shown at the midbody at the time of**

chromatin bridge resolution (Fig. 3F, which is very nice) and lem-3 mutants show longer lived bridges (Fig. 3B and S2, also technically beautiful). It is therefore tempting to speculate that LEM-3 cleaves bridges during cytokinesis. However, stable bridges in lem-3 mutants may have alternative causes. Lack of LEM-3 in mitosis or in the previous S-phase could cause changes in the quantity or quality of bridge DNA, making these bridges more stable and/or causing abscission to be inhibited. Thus, midbody recruitment of LEM-3 and bridge resolution may be, in principle, functionally unrelated events. Unless direct evidence is provided (such as inactivation of LEM-3 specifically during cytokinesis), the authors should make clear in the title, abstract and main text that the evidence supporting LEM-3 dependent cleavage of bridges at the midbody is indirect and that alternative interpretations cannot be excluded”.

Our response: As noted above, we now include new data in Figures 3F and S3B to indicate that midbody function is important to resolve chromatin bridges during cytokinesis, consistent with our data implicating LEM-3 (which is tethered to the midbody and is important for bridge resolution). Furthermore, we agree that LEM-3 might also act outwith the cytokinesis stage and we add a sentence to this effect into the discussion (page 10, lines 26-28). It would be nice to inactivate LEM-3 specifically during cytokinesis, but this not possible technically at present (it would either need a temperature sensitive allele of *lem-3* that can be inactivated within less than three minutes or a specific chemical inhibitor), and is thus beyond the remit of this manuscript.

2. “The authors claim that LEM-3 function at the midbody is important to preserve genome integrity (Discussion, 1st paragraph and also later). Again, no direct evidence supports the claim that LEM-3 functions during cytokinesis. Even if LEM-3 preserves genome integrity (which is an inference), it could do so through a function in previous cell cycle stages, not necessarily depending on its midbody localization during cytokinesis.”

Please see our response above to points 1 and 2.

3. “As noted in the Discussion, nuclease recruitment to chromatin bridges might be deleterious to genome stability, by causing chromosome fragmentation. The authors should comment on how they think LEM-3 processing of chromatin bridges could differ from other nucleases such as TREX1 to preserve genome integrity.”

Our response: Maciejowski et al., (2015) showed that the chromatin bridges induced by telomere fusion can be attacked by the cytoplasmic 3' exonuclease TREX1 and generate extensive RPA-coated single stranded DNA. The TREX1-mediated processing contributes to the resolution of the chromatin bridges. However, the subsequent repair of these bridges (induced by telomere to telomere fusion) leads to the random joining of chromosomes, chromosomal fusions being broken apart in the next cell cycle. All together this can result in broken chromosomes, micronuclei formation and chromothripsis, a phenomenon characterized by the massive localized fragmentation and random reassembly of chromosomal fragments. Therefore, TREX-1 processing is a pathological event, which allows completion of cytokinesis in the first cycle, but this processing is deleterious and leads to genome instability and lethality. LEM-3, in contrast helps to resolve chromatin bridges such that genome integrity is maintained, and we have strong evidence to support this notion. Firstly, when MCM-7 or CAPG-1, which are both essential for viability, are partially depleted by treating with 2% of *mcm-7* RNAi and 20% of *capg-1* RNAi respectively, 80% of wild type worms survive, while the viability is almost completely lost in *lem-3* mutants (Figures 3C and 3D). The hypersensitivity of *lem-3* mutants to the partial inhibition of MCM-7 and CPAG-1 indicates that LEM-3 is able to process DNA intermediates resulting from incomplete DNA replication or partial chromosome decondensation to ensure viability. Secondly, unlike TREX1, LEM-3/Ankle1 showed endonuclease activity (Dittrich et al., 2012 and Brachner et al., 2012) and our work strongly supports that LEM-3 mainly acts on the chromatin bridges passing through the midbody.

4. “In the specific case of unreplicated bridges, the authors may want to discuss their data in the context of a DNA resolution model proposed by Julian Blow and collaborators (Moreno et al., PNAS 2016, doi: 10.1073/pnas.1603252113).”

Our response: Our colleagues in Dundee, Moreno et al., (2016) suggested that in organisms with large genome, unreplicated DNA is frequently present in normal cells at the end of G2 phase. The postreplicative resolution of unreplicated DNA stretches leads to ‘ultrafine DNA bridges’ structures, which are not chromatinized and cannot be stained by conventional DNA dyes. It is possible that the ultrafine DNA bridges exist in early *C. elegans* embryos where rapid cell divisions occur. Therefore, the accumulation of LEM-3 at the midbody during late stages of cell division may also contribute to the resolution of the ultrafine DNA bridges caused by under-replication. We have not directly observed such bridges, which in mammalian cells can be visualized by staining with antibodies against the BLM and PICH helicases. The former is orthologous to *C. elegans* HIM-6, which we did not find associated with bridges (our unpublished data). PICH1 is not conserved in nematodes. Irrespective, yes what Moreno et al discussed is a model that would allow for the error free completion of replication in the G1 phase. To allow for this to happen unreplicated double strands have to be disentangled to allow for the ordered segregation of single stranded DNA. It is well possible that LEM-3 has a role in this process. We have amended the discussion of our results such that the above is much clearer discussed in the revised manuscript (page 11, lines 3-16).

5. “More thorough quantification of the observed phenotypes would strengthen the observed correlations between bridge resolution and LEM-3 function. For instance: In Fig 3B, S2, 5D, how much longer do LEM-3 deficient bridges persist, compared to controls? In Fig S4B, by how much is nuclear separation advanced in lem-3 mutants? In Fig. 1E-F, is advanced recruitment of LEM-3 to the midbody statistically significant?”

Our response: We define persistent chromatin bridges, as linkages that persist into the second cell divisions. The nuclear separation in Figure S4B (now Figure S5A) is calculated by measuring the distance between two separating nuclei. We observed that the two nuclei were far more separated in *lem-3* mutants 200 seconds after anaphase. The average nuclear separation rate in *lem-3* is 5.31 $\mu\text{m}/100\text{s}$ (standard deviation SD, ± 0.13 ; $n=5$), compare to 4.02 $\mu\text{m}/100\text{s}$ in wild type (SD, ± 0.32 ; $n=6$, $p < 0.0001$), 3.77 $\mu\text{m}/100\text{s}$ in *slx-4* mutants (SD, ± 0.23 ; $n=6$, $p < 0.00001$), and 2.4 $\mu\text{m}/100\text{s}$ in *lem-3*; *slx-4* double mutants (SD, ± 0.37 ; $n=5$, $p < 0.00001$). This data is now included in the manuscript.

Also a statistical analysis of the intensity and the advance recruitment of LEM-3 to the midbody in the presence of chromatin bridges induced by *mcm-7* or *capg-1* RNAi is provided (Figure 1E-H).

6. “The midbody localization of LEM-3 requires the central spindle and Aurora-B kinase function (Figure 4). Is a midbody formed in the absence of AIR-2 function? If not, the role of AIR-2 in LEM-3 localization in these experiments could be indirect. This should be noted. To directly test the role of Aurora-B in LEM-3 midbody recruitment, the authors could inactivate AIR-2 after anaphase onset, using an *air-2* *ts* mutant such as *air-2(or207)*.”

Our response: Midbody function depends on AIR-2 (Schumacher et al., 1998). However, our finding of the colocalization of LEM-3 and AIR-2 at the midbody (Figure 4A), LEM-3 containing conserved AIR-2/Aurora B consensus phosphorylation sites (K/R; K/R; X0-2; S/T)(Figure 4C), these sites being phosphorylated in vivo (Figures 6A, 6B and S7), and mutation of these sites abrogating LEM-3 midbody localization and bridge processing (Figure 5C-E), all together provides strong evidence for AIR-2 being directly involved in the regulation of LEM-3. Yes mechanistic details will have to be further worked out and this would include taking advantage of rapid temperature shift experiments using an *air-2* *ts* allele. At present we do not have access to the required special microscopy stages, and this together with us not having the required strain including the necessary transgenes at hand, is why we

decided to not focus on this temperature shift experiment. This experiment in the context of this manuscript would provide an incremental but not a major advance. Overall, our data strongly indicate that AIR-2 is involved in the regulation of LEM-3. We elaborate on this in the discussion (page 10, lines 9-15) mindfully that at the present stage we did not fully work out if AIR-2 directly regulates LEM-3.

7. "The Aurora B kinase dependent NoCut checkpoint is essential to detect chromatin bridges and prevent abscission until chromatin bridges have been properly resolved" (p. 6, bottom). At least in budding yeast, NoCut delays abscission but does not prevent resolution of decondensed and catenated bridges (Amaral et al., Nat Cell Biol 2016, doi: 10.1038/ncb3343)."

Our response: We agree. What we want to state is that the Aurora B kinase dependent NoCut checkpoint is able to detect chromatin bridges and delay abscission, allowing time for chromatin bridge resolution. We have adjusted the text to make this clearer (page 8, line 2-3).

8. "The localizations of LEM-3 and AIR-2 at the midbody seem mutually exclusive in some images (Fig 4A). This should be quantified."

Our response: We observed that LEM-3 did not fully colocalize with AIR-2 at the midbody. We now included quantitative data (page 8, lines 3-6) about this using the Mander's overlap coefficient (MOC), revealing that 77.4% (standard deviation, $\pm 9.3\%$; n=12) of the LEM-3 protein colocalized with AIR-2 at the midzone/midbody.

9. "LEM-3 localizes as a broad band bisecting the segregating chromosomes in Figure 1C. Could this be the contractile ring, or the spindle midzone microtubules? This localization is not seen in other figures, where LEM-3 only is visible at the midbody. Is this due to differences in imaging techniques?"

Our response: OK, clearly the static images in Figure 1C acquired by super-resolution microscopy provide higher resolution as compared to real time imaging by spinning disc microscopy. However static images cannot provide exact temporal resolution. Indeed, LEM-3 always congresses to the midbody. However, when bridges are induced by underreplication LEM-3 initially occurs in a broader band. Also upon condensin depletion this is the case but LEM-3 localization initially focused on threads, probably due to less chromatin bridges induced between segregating nuclei (Figure 6B). Thus congression into the midbody is gradual. To investigate LEM-3 localization dependency on the contractile ring we depleted the NMY-2 contractile ring component. Depletion of NMY-2 inhibited furrow formation and ingression but didn't affect the midzone formation (Fig. S3A). We found that LEM-3 could still be detected at the midzone upon *nmy-2* RNAi (Fig. S3A), further strengthening that LEM-3 is specifically recruited to the midzone region (Figures 1D and 2A).

10. "Unlike LEM-3, TREX1 associates throughout chromatin bridges and not just with the region at the midbody" (p. 9). However, LEM-3 also seems to associate with chromatin bridges during anaphase (see Fig. 6B)."

Our response: We found that LEM-3 was first detectable at the midzone. As cytokinesis proceeds, LEM-3 further accumulates and its localization becomes more restricted to the midbody (Figure S2B, white arrow; Movie S5). In contrast, TREX1 has been reported to be present throughout the chromatin bridges and no specific accumulation of TREX1 at the midbody could be observed (Maciejowski et al., 2015). In the revised manuscript our wording is more precise (page 11, lines 21-23).

Reviewer #2 (Remarks to the Author):

The reviewer summarized her/his view as follows: **"This manuscript makes a convincing case that LEM-3 acts to promote accurate chromosome segregation in the early *C. elegans* embryo. The authors present**

evidence for enhancement of a number of apparent null alleles in DNA repair pathways, consistent with the idea that it acts in a novel repair pathway. The authors also convincingly demonstrate that LEM-3 accumulates at the midbody and they suggest that it functions at the location. However, the evidence for this latter conclusion is not convincing and further analysis along these lines is warranted prior to publication. This also renders the title problematic.”

Specific points:

1. **“The central conclusion of the manuscript that LEM-3 acts at the midbody is based on LEM-3 protein localization and the dependence of this localization on centralspindlin and Aurora B. This conclusion has several problems. First, no evidence is shown that the ability of LEM-3 to resolve chromosome bridges requires an intact spindle midzone and LEM-3 recruitment to that site. This could be easily tested by combining some of the weak DNA damage perturbations with ZEN-4 depletion and scoring for an enhancement of the frequency of chromosome bridges.”**

Our response: We thank the reviewer for this helpful suggestion, and now show that depletion of the central spindle components ZEN-4 or CYK-4, which are required for localization of LEM-3 during cytokinesis (already shown for *cyk-4* RNAi in Figure 2C of the original submission), leads to increased frequency of chromosome bridges. As shown in Figures 3F and S3B, embryos solely treated with low dose of *mcm-7* RNAi, or with RNAi against *zen-4* or *cyk-4*, never showed chromatin bridges. In contrast, when embryos were treated with a combination of partial *mcm-7* RNAi and *zen-4* or *cyk-4* RNAi, chromatin bridges formed with a high penetrance in (4/7) and (4/9) cases, respectively (Fig. 3F and S3B, Movies S6 and S7). These results demonstrate that midzone assembly is required for chromatin bridge resolution, as well as for LEM-3 recruitment and cytokinesis completion.

2. **“Second, on page 6, the authors state ” In addition, LEM-3 localization depends on the AIR-2/Aurora B Kinase (Fig. 4B), indicating that AIR-2 regulates the midbody association and/or activity of LEM-3”. Given that**

AIR-2 is required for spindle midzone assembly and the previously demonstrated dependence of LEM-3 localization on ZEN-4 (though not necessarily function - see above), this is a trivial result.”

Our response: We take the reviewer’s point, but also show that mutation of candidate AIR-2 phosphorylation sites in LEM-3 compromises localization of LEM-3 at the midbody.

3. “In figure 1A, untreated embryos accumulate LEM-3 at midbodies independent of DNA damage. This should be explicitly stated. Interestingly, this accumulation appears to depend on S192S194 phosphorylation and catalytic activity (Figure 5). While this suggests it is not strictly non specific, given that it is damage independent, it is not clear why the mutations in the catalytic site would impair this damage independent localization. How do the authors explain this result? Does the p-LEM-3 antibody detect the epitope in control embryos that do not express GFP-LEM-3?”

Our response: Indeed, we acknowledge that LEM-3 also associates at the midbody without perturbing DNA metabolism, albeit midbody association is enhanced when (visible) bridges occur (Figures 1D and 2A) Thus initial LEM-3 targeting to the midbody may be independent of chromatin bridge formation. Alternatively, of course, as discussed in the response to reviewer 1, midbody localization of LEM-3 could be required for resolution of ultrafine DNA bridges, which might occur in untreated normal embryos. In fact, it was surprising to us that the double mutant, which is predicted to be catalytically inactive also fails to localize to the midbody. We have validated this result using a second strain independently generated by CRISPR/CAS-9 genome editing. We speculate that those mutated residues may directly or indirectly affect DNA binding and or midbody localization. Yes in the supplements we show that the antibodies are specific (Figure S7A-C). In summary, we now make it more explicit in the text (page 3, lines 28-30) that the basal level of LEM-3 midbody localization occurs without inflicting perturbation in DNA metabolism and are also more explicit about the catalytic site mutations, saying that the mutants we

generated may also affect DNA binding and or midbody localization (page 9, lines 16-19).

4. “Page 4: LEM-3 “dynamically” localizes to the midzone. The data does not show dynamic localization. This would require FRAP or related technique to show protein dynamics. Regulated accumulation is more accurate.”

Our response: We used the word ‘dynamically’ to indicate the gradual accumulation of LEM-3 at the midzone. As cytokinesis proceeds, LEM-3 concentrates and its localization becomes more restricted to the midbody. Following the reviewer who pointed out that the word ‘dynamic’ is more commonly used to describe the turnover of proteins, we now refrain from using the word in the revised manuscript.

5. “The evidence that Aurora B phosphorylation promotes LEM-3 function is not highly convincing. The difference in radiation resistance is not particularly impaired relative to the wild-type, it only mildly enhances the inviability in combination with slx-4 mutation. The part of the manuscript might be better de-emphasized.”

Our response: We agree that the *lem-3 S192A S194A* phosphosite mutant didn’t show phenotypes as severe as the *lem-3* null mutant and the mutant with mutations in the conserved nuclease domain, indicating LEM-3 function is impaired but not abolished by mutation of these two conserved serines. This may be due to the residual midbody location of this phosphosite mutant. We also cannot exclude that there are additional AIR-2 dependent phosphorylation sites on LEM-3. We now discuss this better in the Discussion on page 10 lines 17-21.

6. “The authors use operational terms for the extent of RNAi mediated depletion, corresponding to dilution of the bacteria used for RNAi. Given that there is no quantification of extent of depletion, these numbers are

meaningless to the reader. They should use qualitative terms like partial, or weak.”

Our response: Yes, the percentage we used in our study indicates proportion of bacteria expressing *mcm-7* RNAi or *capg-1* RNAi. We have explained it in the text and used terms such as partial, weak or low dose to show the extent of RNAi.

7. “In some of the experiments (see page 4), the number of the analyzed embryos is on the low side. 10 embryos is more frequently used in the field.”

Our response: As suggested by the reviewer, we have increased the number of the analyzed embryos by repeating the experiments of weak *mcm-7* RNAi for both wild type and *lem-3* mutant worms, the result (Figure 3A) showed that chromatin bridges only formed in *lem-3* mutant (sample size n=12) but not in wild type worms (n=10) when treated with low dose of *mcm-7* RNAi, which is consistent with our observation in the study.

8. “Figure preparation: Where appropriate, all datapoints should be shown in bar graphs, e.g. Fig E-H. It is also convention that embryos are oriented with the anterior to the left, not vertically.”

Our response: We have changed all figures as suggested so that the embryos are oriented from vertically to horizontally.

9. “The authors should point out in the main text that the *him-18*, *hrc-1*, *brc-1*, and *lem-3* alleles are nulls. This is critical for the conclusion that *LEM-3* acts in a distinct pathway.”

Our response: Thanks, we now clearly point this out. The *lem-3* allele (*mn155*) we used in the study has a premature stop codon at R190 and this mutation leads to a truncated protein lacking the LEM domain and the nuclease domain (Dittrich et al., 2012). In addition, it fails to complement

another *lem-3* allele (*op444*). The other mutants namely *him-18* (*tm2181*, Saito et al., 2009), *brc-1* (*tm1145*, Adamo et al., 2008), *lig-4* (*ok716*, Lemmens et al., 2013) and *polq-1* (*tm2026*, Koole et al., 2016) have all been previously shown to be null alleles.

10. ***“Page 2, “failure to do so can lead to the severing of chromosomes, prevent cell division or cause aneuploidy and polyploidization 1-4, molecular pathologies that all result in disease.” This is stated too strongly and is therefore incorrect. Many chromosome segregation defects result in death of the affect cell and no subsequent disease. Certainly there are other cases that are disease-associated. In addition, the processes that affect repair of some chromosome segregation defects can improve situation but they do not necessarily ensure euploidy. For example, post anaphase, perhaps the repair events prevent chromosome breakage, but the daughter cell genomes are not necessarily intact.”***

Our response: We agree with the reviewer that aneuploidy and polyploidization are hallmarks of some genetic diseases and chromosome segregation defects do not necessarily result in disease. We have changed the sentence (page 2, lines 5-7) and are more careful in our wording.

11. ***“Page 2, “ next to nothing ”, perhaps little is better.”***

Our response: Thanks, done.

12. ***“Page 4, “If CYK-4 or SPD-1, a protein”. Incorrect tense.”***

Our response: Thanks, we have corrected tense.

13. ***“Page 5, “Whereas 80% of wild type worms survived treatment with 2% of mcm-7 RNAi or 20% of cpag-1 RNAi respectively, either treatment led to a strong synthetic lethal effect in lem-3 mutants (Fig. 3C and 3D).” State the result quantitatively, with more specificity than strong.”***

Our response: Thanks, we have changed the text to state it quantitatively.

14. ***“Page 6, “found that LEM-3 localization commenced “. LEM-3 was first detectable at the midzone at the time of cleavage furrow completion.”***

Our response: Thanks. Done.

15. ***“Page 6, “The corresponding sites are known to be”. Awkward phrasing.”***

Our response: We have reworded the sentence.

16. ***“Page 10, last chance ‘saloon’. Inappropriate placement of quotes. Should be “last chance saloon”.***

Our response: Thanks, we have corrected this.

Reviewer #3 (Remarks to the Author):

We were glad to see that the reviewer thought that ***“The authors make a strong case that LEM-3 acts at the midbody of dividing cells to resolve DNA bridges...These results will be interesting to the DNA repair field as they suggest a new mechanism for the resolution of chromatin bridges.”***

The reviewer then went on to raise a number of major and minor concerns, as discussed below:

Major points:

1. ***“Introduction: It would be helpful to describe what is known about LEM-3 and Ankle1 rather than summarizing three prior publications with “Ankle1 is poorly characterized.”***

Our response: Thanks, we have added a short paragraph to describe the known functions of LEM-3 and Ankle1 in the Introduction (page 2, lines 23-30

and page 3, line 1).

2. “The first two sections of results: “LEM-3 accumulates at the midpoint of chromatin bridges during cell division” and “LEM-3 associates with the midbody during cytokinesis” are similar titles, but the former has a more broad characterization of LEM-3 and lem-3 mutants. These sections need to be organized more clearly.”

Our response: We thank the reviewer for this suggestion. We have reorganized the first two sections of Results and make it clearer by using new titles.

3. “The authors state “lem-3 mutants synergize with mutants defective in the three major DNA double-strand break repair modalities,” but it is unclear that this figure shows synergy in the mutants. Synergism in genetic terms means that the effect seen in the double mutant is greater than the combination of the two single mutants (that is, more than just an additive effect). Fig. S1A doesn’t appear to show synergy between brc-1 and lem-3. Although one would expect a synergistic increase when mutations in two different pathways are combined, if the effect is really additive it needn’t argue against their hypothesis. Finally, a “mutant” is an individual, whereas a “mutation” is a change in the genetic material. It is mutations that are showing additive effects (in double-mutant animals).”

Our response: Regarding ‘synergize’, indeed both the first and senior authors are not native English speakers, and we concur that we at times misused the word ‘synergize’, especially when considering the strict genetic definition of this term. What we wanted to say is that *lem-3* mutants showed more severe phenotypes when combined with mutations defective in the three major DNA double-strand break repair modalities after IR treatment, which indicates that LEM-3 probably acts in a previously unknown DNA repair pathway. We have adjusted the text and are now more careful to distinguish between ‘mutant’ and ‘mutation’.

4. ***“Page 4 – “We observed LEM-3 localization at the centre of the midbody slightly after ZEN-4 midbody location became apparent (Fig. 1I; Movie S3).” I do not see clear evidence of LEM-3 appearance after ZEN-4, it appears it is at the same time according to Fig. 1I and Movie S3.”***

Our response: The mCherry-ZEN-4 started to accumulate at the midzone about 100 seconds after anaphase onset and it congressed into a focus at 200 seconds (Figure 2A and Movie S3). Whereas, the signal of YFP-LEM-3 is quite diffused at the midzone 200 seconds after anaphase onset and the distinct LEM-3 foci started to appear 300 seconds after anaphase onset (Figure 2A, 1E and 1F; Movie S3). However we cannot exclude that this could be due to the stronger intensity of mCherry-ZEN-4 compared with the intensity of YFP-LEM-3. We are now very careful with our observation and stated that we observed LEM-3 colocalize with ZEN-4 at midbody position (Figure 2A and Movie S3).

5. ***“Figure 3A. An important control to show here is lem-3 mutants without the 10% mcm-7 RNAi to show that the chromatin bridges are a result of having replication stress, and not just the lem-3 mutation.”***

Our response: Thanks for pointing this out. We indeed did the time-lapse recording of first mitotic division of the *lem-3* mutant without the treatment of 10% *mcm-7* RNAi. Figure S4A and S5C (10/10 embryos) showed that there are no detectable chromatin bridges in the *lem-3* mutant during the first mitotic divisions. We have highlighted this result and made it clear on page 5, lines 15 and 16.

6. ***“Page 6- “separation of nuclei occurs faster in lem-3 mutants, indicating a potential role for LEM-3 in delaying the separation of nuclei (Fig. S4B).” This figure also shows that the lem-3; slx-4 double mutants have a slower separation of nuclei compared to the control, which is the opposite phenotype of the lem-3 single mutants. The authors should comment this observation?”***

Our response: In *lem-3* single mutant, no chromatin bridges could be detected between dividing nuclei (Figure S4A). Indeed we found that the faster separation of nuclei in the absence of LEM-3 (Figure S4A and S4B). This phenotype can also be observed in the mutants lacking the central spindle components such as CKY-4 or ZEN-4 (our unpublished data). However, the separation of nuclei was compromised due to formation of the chromatin linkages between dividing nuclei in *lem-3; slx-4* double mutants, as indicated in Figure S4A and S4D.

7. “Page 6- Figure 3F. The authors say there is an increase of abundance of LEM-3 localization, but this is unclear from the figure and is not quantified. It is also unclear from the text that this is in a *capg-1* RNAi background.”

Our response: We found that LEM-3 was first detectable at the midzone at the time of cleavage furrow completion, then increased in abundance and together with the chromatin bridges congressed into the midbody (Figure S2B, white arrow; Movie S5). We now include a quantitative analysis of the intensity of LEM-3 in Figure S2C, which shows accumulation of LEM-3 at the midzone after anaphase onset. We also make it clear that the chromatin bridges were induced by *capg-1* RNAi on page 7 lines 10-13.

Minor points

1. “Figure 1C- include a label showing this is *mcm-7* RNAi. Are these pictures from a time course? If so, label it as such.”

Our response: These pictures are from DAPI stained different stages of embryos treated by *mcm-7* RNAi. We have indicated this in the main text on page 4, lines 6-9.

2. “Figure 2A. In the *mcm-7* & *cyk-4* mutant, are the lines between the nuclei supposed to represent chromatin bridges? Because they look very similar to the central spindle in the control. Perhaps color the DNA, so it is clear that those lines are DNA and not the central spindle.”

Our response: Thanks, we have colored the DNA and made it clear that the blue lines between the nuclei in the *mcm-7* & *cyk-4* RNAi embryo represent chromatin bridges (Figure 2B).

3. ***“Figure 1E-H are never referenced to in the text. This could belong with the first paragraph of the section, “LEM-3 is required for the resolution of chromatin bridges” pages 4/5.”***

Our response: Thanks, we have adjusted the text and described the result in the section “LEM-3 is required for the resolution of chromatin bridges” on page 5 lines 6-9.

4. ***“Page 9- second paragraph of Discussion. Could also note that the phosphosite mutant is not as affected by IR as a lem-3 null, suggesting that the residual midbody localization of the phosphosite mutant could be partially rescuing this phenotype.”***

Our response: We fully agree that the *lem-3 S192A S194A* phosphosite mutant is not as sensitive to IR as the *lem-3* null mutant, indicating that mutation of these two serines impairs but does not completely abolish LEM-3 function. This may be due to the residual midbody location of this phosphosite mutant. We now highlight this in the Discussion on page 10 lines 17-21.

REVIEWERS' COMMENTS:

Reviewer #1 (Remarks to the Author):

The authors have addressed most of my concerns and this version is considerably improved. The most important addition is the demonstration that double RNAi of zen-4 and mcm-7 leads to chromatin bridge formation. This finding strengthens the hypothesis that midbody-associated LEM-3 contributes to the correct resolution of DNA during cytokinesis. As mentioned below, a more accurate interpretation of these new results is that the spindle midzone prevents the formation of bridges during anaphase and possibly cytokinesis. That being said, a few points from my previous review remain to be clarified.

1. LEM-3 action during cytokinesis is still not formally demonstrated, although the new data in Figure 3F add support to this hypothesis. I would recommend caution interpreting this data, however – in particular, an alternative interpretation is that LEM-3 acts during both anaphase and cytokinesis. This takes into account that (a) LEM-3 localizes not only to the midbody during cytokinesis but also to the midzone in anaphase; (b) early anaphase bridges are not observed in mcm-7 RNAi but are present in mcm-7 zen-4 double, suggesting LEM-3 already starts to act during anaphase. Therefore, and to avoid misleading statements, it would be correct to replace “during cytokinesis” with “during late mitosis” or similar wording in the title, abstract and main text.

2. The authors acknowledge in their rebuttal letter that Aurora B is required for midzone function, yet this is not reflected in the manuscript. While I agree that there is additional evidence suggesting that Aurora-B regulates LEM-3 localisation and function, the fact remains that Figure 4A-B does not support the conclusion that Aurora “is required for LEM-3 association with the midbody” except in a very indirect manner. The statement “we have no evidence for the direct interaction...” (Discussion, 2nd paragraph) is not sufficient without mentioning that Aurora-B is required for midzone assembly. The authors should clearly mention the caveats associated with the interpretations of their experiments!

3. Please clarify that “persistent bridges” are defined (according to the rebuttal letter) as bridges that persist until the next division.

Reviewer #2 (Remarks to the Author):

This manuscript is greatly improved and nearly ready for publication.

There are three lingering points:

1- Figure 4. Figure 4B remains problematic. As several reviewers pointed out, since AIR-2 is required for ZEN-4 accumulation at the midzone, and ZEN-4 is required for LEM-3 localization, it is a trivial result that AIR-2 depletion prevents LEM-3 localization.

Second, while it is true that both serine residues in Fig 4C can be phosphorylated by Aurora B in vitro, the first one is a better substrate and more recent evidence suggests that the more c-terminal site is phosphorylated by Ndr kinases not aurora B. Perhaps 4A could be included in figure 2.

2 - The bar graphs should be replaced with graphs that reveal the actual data points, with the average values indicated. See this link for example:

<https://zenodo.org/record/574883#.WguGWUyZMUE>

3 - Minor point, the word "worm" is vague and colloquial. The authors should use *C. elegans* or nematode instead.

Reviewer #3 (Remarks to the Author):

This is a revision of a previously reviewed manuscript that describes a role for *C. elegans* LEM-3 nuclease in resolving chromatin bridges during cytokinesis. The authors have conscientiously responded to criticisms from all three reviewers. I have no remaining criticisms.

Response to reviewers

As suggested by reviewer 1 we now provide a more nuanced discussion of our data clearly stating that we think that LEM-3 also acts at the midzone during anaphase in addition to the role at the midbody in cytokinesis. Accordingly we changed wording in our title and abstract replacing during 'cytokinesis' with 'late mitosis'.

Also, following suggestions by reviewers 1 and 2 we have tuned down our wording on the regulation of LEM-3 by AIR-2, more clearly stating that this could be indirect via the AIR-2 dependent assembly of the midzone. Taking on the argument of the reviewers we now shifted the AIR-2 RNAi depletion experiment (Fig. 4a) to the Supplementary section. Having this done allowed for space to take two datasets into the main Figures. One (Fig. 3c) demonstrates LEM-3 dependent bridge resolution upon CAPG1 depletion. The second one (Fig. 3g) shows the synergy of *lem-3* and *brc-1* (*C. elegans* *brca1*). We think that this data is very important and should be visible, given the inferred role of the LEM-3 homolog Ankle1 in affecting the incidence and severity of breast and ovarian cancers, including those associated with by *brca1* deficiency.

As per reviewer 1 we clarified how we define persistent bridges in the text. Also following reviewer 2 we changed the bar graphs in Fig. 1 to the format suggested, which better represents the data. Finally, we also changed 'worm' to 'nematode' or '*C. elegans*'.